# Protein-altering variants at copy number-variable regions influence diverse human phenotypes

Margaux L. A. Hujoel [1,2,3] ✉, Robert E. Handsaker [3,4,5],
Maxwell A. Sherman [1,2,3,6,11], Nolan Kamitaki [1,2,3,7], Alison R. Barton [1,2,7,12],
Ronen E. Mukamel[1,2,3], Chikashi Terao [8,9,10], Steven A. McCarroll[3,4,5] &
Po-Ru Loh [1,2,3] ✉

Copy number variants (CNVs) are among the largest genetic variants, yet CNVs have not been effectively ascertained in most genetic association studies. Here we ascertained protein-altering CNVs from UK Biobank whole-exome sequencing data (*n* = 468,570) using haplotype-informed methods capable of detecting subexonic CNVs and variation within segmental duplications. Incorporating CNVs into analyses of rare variants predicted to cause gene loss of function (LOF) identified 100 associations of predicted LOF variants with 41 quantitative traits. A low-frequency partial deletion of *RGL3* exon 6 conferred one of the strongest protective effects of gene LOF on hypertension risk (odds ratio = 0.86 (0.82–0.90)). Protein-coding variation in rapidly evolving gene families within segmental duplications—previously invisible to most analysis methods—generated some of the human genome's largest contributions to variation in type 2 diabetes risk, chronotype and blood cell traits. These results illustrate the potential for new genetic insights from genomic variation that has escaped large-scale analysis to date.

Genomic structural variants (SVs), which modify from 50 base pairs to megabases of DNA, account for most base pairs of variation in each human genome[1]. Recent major efforts to study structural variation in human genomes elucidated the landscape and mutational origins of SVs by ascertaining SVs from short-read sequencing of many thousands of individuals[2,3] and long-read sequencing of tens of individuals[4,5].

Copy number variants (CNVs) are an important class of SVs with unique functional consequences (for example, by modifying the dosage of genes or regulatory elements). Assessing the impact of CNVs on human phenotypes requires genotyping CNVs in large well-phenotyped cohorts. This has been possible for larger CNVs detectable from the SNP array and whole-exome sequencing (WES) data generated at

[1]Division of Genetics, Department of Medicine, Brigham and Women's Hospital and Harvard Medical School, Boston, MA, USA. [2]Center for Data Sciences, Brigham and Women's Hospital and Harvard Medical School, Boston, MA, USA. [3]Program in Medical and Population Genetics, Broad Institute of MIT and Harvard, Cambridge, MA, USA. [4]Stanley Center for Psychiatric Research, Broad Institute of MIT and Harvard, Boston, MA, USA. [5]Department of Genetics, Harvard Medical School, Boston, MA, USA. [6]Computer Science and Artificial Intelligence Laboratory, Massachusetts Institute of Technology, Cambridge, MA, USA. [7]Department of Biomedical Informatics, Harvard Medical School, Boston, MA, USA. [8]Laboratory for Statistical and Translational Genetics, RIKEN Center for Integrative Medical Sciences, Yokohama, Japan. [9]Clinical Research Center, Shizuoka General Hospital, Shizuoka, Japan. [10]Department of Applied Genetics, School of Pharmaceutical Sciences, University of Shizuoka, Shizuoka, Japan. [11]Present address: Serinus Biosciences Inc., New York, NY, USA. [12]Present address: Department of Human Evolutionary Biology, Harvard University, Cambridge, MA, USA. ✉e-mail: mhujoel@broadinstitute.org; poruloh@broadinstitute.org

scale by biobank projects and consortia[6–11]. However, the effects of kilobase-scale and smaller CNVs, which comprise most CNVs[1,5], have remained largely hidden, requiring analyses of whole-genome sequencing (WGS) datasets[12,13]. Such analyses demonstrated the important influences of CNVs (and other SVs) on gene expression[14,15] but have only recently begun reaching the scale necessary to detect associations with human phenotypes[16–19].

We sought to leverage population genetic principles to address this challenge for protein-altering CNVs. Studies of CNVs classically focused on large, extremely rare CNVs that recurred ab initio in different individuals or families[20]; most such CNVs affected many genes, making it hard to discern the mechanism according to which they affected phenotypes. In contrast, far more CNVs are inherited by many people from common ancestors; these CNVs, which are generally smaller but can have disabling effects on specific, individual genes (and thus interpretable, specific effects on human biology), have often gone undetected. As such CNVs are inherited by descent from common ancestors, we hypothesized that the additional information provided by SNP haplotypes[9,21] could enable the analyses of abundant exome sequencing data to detect even small copy number-altering CNVs within individual protein-coding genes, including genes within multicopy and segmental duplication regions. We applied this approach to explore the impacts of protein-altering CNVs on the approximately 500,000 research participants in the UK Biobank (UKB)[22,23].

## Results

### Haplotype-informed detection of rare protein-altering CNVs

We first sought to sensitively detect rare protein-altering CNVs of all sizes, including CNVs that affected single exons, from UKB exome sequencing data ($n = 468,570$). To enable detection at this resolution, we used a computational approach that can integrate information across individuals who share extended SNP haplotypes (Fig. 1a). Because CNVs inherited according to descent from common ancestors tend to be inherited on a shared SNP haplotype, analyzing such individuals together increases detection sensitivity (Fig. 1a). We previously used this approach to detect CNVs from genotyping array intensity data (while retaining sensitivity to larger de novo CNVs)[9]; in this study, we adapted the approach to model exome sequencing read counts using negative binomial distributions with sample-specific and region-specific parameters (Methods). Importantly, leveraging haplotype-sharing information enabled analysis at 100-bp resolution, allowing detection of small CNVs that only partially overlap single exons (Fig. 1a).

We applied this approach to identify CNVs in the full UKB cohort, focusing our main analyses on 454,682 participants with European ancestry to avoid confounding in subsequent association analyses. We identified an average of 93.4 CNVs per individual (65.7 deletions and 27.7 duplications), roughly half of which were short deletions called across intervals of 500 bp or less (Fig. 1b and Supplementary Table 1). This represented a twofold increase compared to a recent analysis of an interim UKB WES release ($n = 200,000$)[10]. Validation using WGS data for 100 participants indicated that false positives were well controlled at less than 10%, with precision improving modestly with CNV size (Fig. 1c, Supplementary Table 2 and Methods). Most deletions and roughly half of duplications affected at most one exon (Fig. 1d), including some CNVs identified using only off-target reads that did not intersect any exons.

The most impactful variants were uncommon: across 18,651 genes, whole-gene duplications and CNVs predicted to cause loss of function (pLOF) were identified in a median of eight and 11 individuals, respectively, with observed counts decreasing with increasing gene constraint (Fig. 1e). When focusing on genes rarely altered by such events, a mean of 4.4 genes per individual were affected (1.8 genes for whole-gene duplications and 2.6 genes for pLOF CNVs) (Fig. 1f), indicating improved sensitivity compared to state-of-the-art methods for rare CNV detection[24].

### Rare large-effect CNVs implicate new gene–trait relationships

This resource of protein-altering copy number variation in the UKB made it possible to discover new links between genetic variation and human phenotypes. To do so, we analyzed CNVs for association with 57 heritable quantitative traits (reflecting a broad spectrum of biological processes; Supplementary Data 1) using linear mixed models[25,26]. We performed two sets of association analyses (Extended Data Fig. 1): (1) CNV-only analyses, which identified 180 CNV–trait associations ($P < 5 \times 10^{-8}$) probably driven by nonsyndromic CNVs (Fig. 2a and Supplementary Data 2); and (2) gene-level burden analyses that collapsed all types of pLOF variants (nonsyndromic CNVs, single-nucleotide variants (SNVs) and indels) to maximize power to detect rare LOF effects. The burden analyses identified 100 pLOF gene–trait associations ($P < 5 \times 10^{-8}$) undetectable from the analyses of pLOF SNVs and indels alone, demonstrating the benefit of incorporating CNVs in burden analyses (+20% increase in associations; Fig. 2b and Supplementary Data 3).

Several of these associations implicated new gene–trait relationships, even for well-studied traits such as height for which common variant association studies have reached saturation[27]. These included strong height-reducing effects (>1 s.d.) of ultrarare pLOF variants (combined allele frequency (AF) < 0.0001) in *CHSY1*, which encodes an enzyme that synthesizes chondroitin sulfate (a structural component of cartilage), *UHRF1*, which encodes an E3 ubiquitin ligase that shares structural homology with UHRF2 (recently implicated by our previous work[9]) and *CDK6*, which harbors one of the strongest common variant associations with height[28]. Rare pLOF variants in two other genes exhibited moderate height-reducing effects (−0.5 s.d.): *USP14*, which encodes a ubiquitin-specific protease and *PRKG2*, which was recently implicated in autosomal recessive acromesomelic dysplasia[29].

Another height association only discovered using CNVs involved *CCNF*, at which a rare duplication spanning a single 107-bp exon accounted for more pLOF events than all other CNVs, SNVs and indels combined (Fig. 2c). Validation using available UKB WGS data ($n = 200,000$) confirmed this CNV as a tandem duplication that was called from WES with 100% precision and 95% recall, illustrating the efficacy of haplotype-informed CNV detection (Extended Data Fig. 2a,b and Supplementary Note). *CCNF* pLOF CNVs associated with a moderate decrease in height (−0.4 ± 0.1 s.d., $P = 5.2 \times 10^{-12}$) and appeared to have a pleiotropic effect on erythrocyte traits (Fig. 2d), motivating further study of this gene and its product, cyclin F.

While further work will be needed to confirm these findings and establish causality, two additional analyses provided evidence supporting their robustness. First, across the 15 height associations discovered only when considering pLOF CNVs, the effect sizes of pLOF CNVs exhibited broad consistency with those of pLOF SNVs and indels (Fig. 2e); this consistency held across traits (Extended Data Fig. 3). Second, for seven height-associated pLOF CNVs that affected genes not previously identified either by large-scale pLOF SNV/indel burden analyses[23,30] or CNV analyses[9], we attempted replication in the BioBank Japan (BBJ)[31], observing broadly consistent effect sizes for the five genes with at least five pLOF CNV carriers in the BBJ (Fig. 2f).

### *RGL3* LOF is associated with reduced hypertension risk

A low-frequency (AF = 0.9%) deletion of part of exon 6 of the *RGL3* gene was associated with lower blood pressure (BP) (−0.11 ± 0.01 s.d.; $P = 6.1 \times 10^{-23}$) and decreased hypertension risk (odds ratio (OR) = 0.86 (0.82–0.90); Fig. 3a,b and Supplementary Table 3) as well as decreased serum calcium (−0.08 ± 0.01 s.d.; $P = 6.0 \times 10^{-11}$; Supplementary Data 2). Closer examination of this CNV showed it to be a 1.1-kb deletion present in 8,117 UKB participants that intersects only 55 bp of coding sequence (Fig. 3c and Extended Data Fig. 2c), yet had been successfully called with 99.9% precision and 88% recall (based on breakpoint-based follow-up analysis; Supplementary Note). This association was replicated in the *All of Us* (AoU) cohort ($n = 245,394$) with a consistent

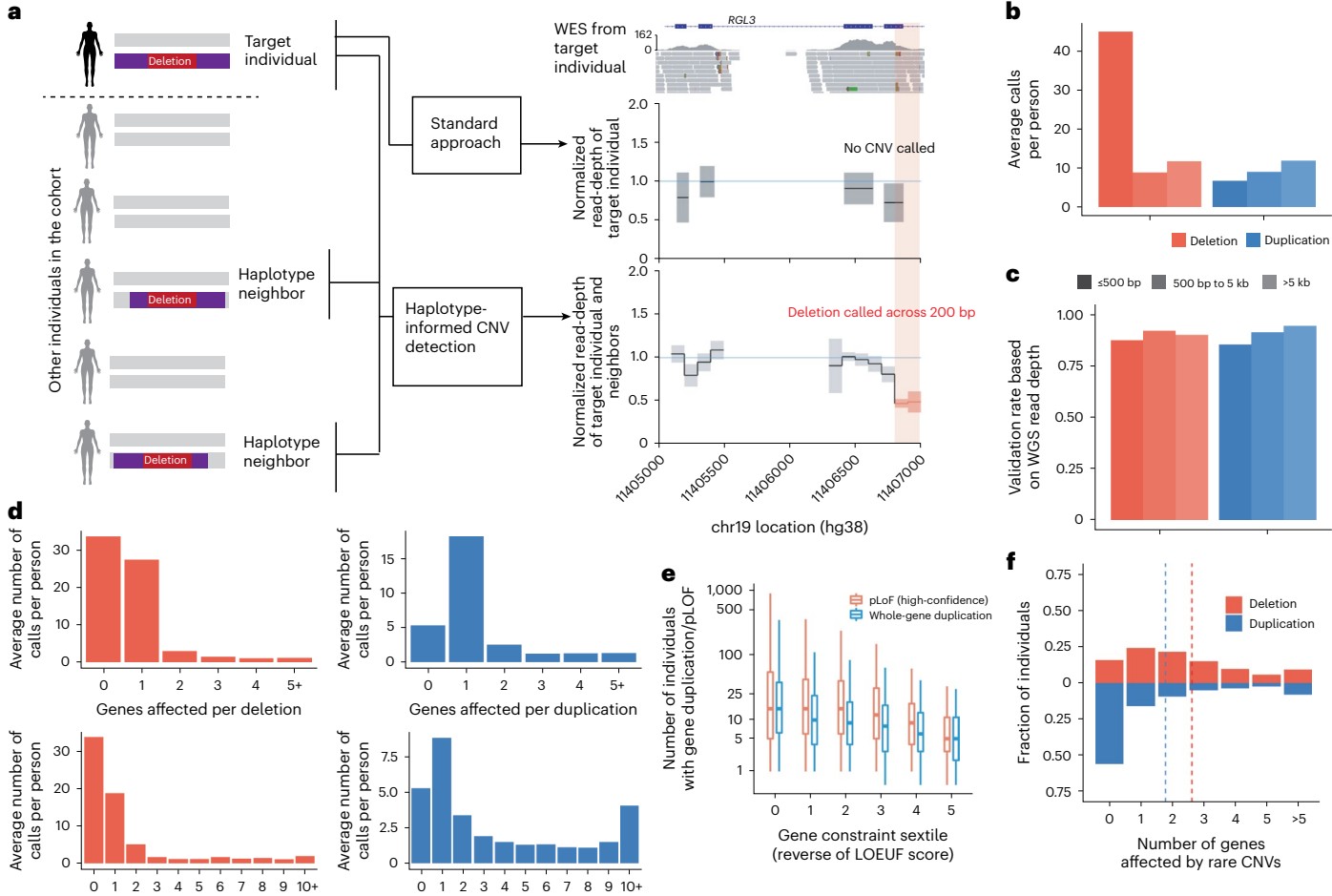

**Fig. 1 | Haplotype-informed CNV detection from UKB whole-exome sequencing data. a**, This approach improved the power to detect CNVs by analyzing WES read-depth data from an individual together with the corresponding data from individuals sharing extended SNP haplotypes ('haplotype neighbors'), facilitating analysis at a resolution of 100-bp bins. In contrast, standard approaches analyze data from an individual alone, generally at exon-level resolution. **b**, Average number of CNVs called per UKB participant, subdivided according to copy number change (deletion or duplication) and call length. **c**, Validation rate of CNV calls based on the analysis of WGS data for 100 UKB participants. **d**, Average numbers of CNVs called per UKB participant affecting the given numbers of genes or exons. **e**, Distributions (across increasingly constrained gene sets) of observed counts of pLOF deletions and whole-gene duplications in 487,205 UKB participants. LOEUF, LOF observed/expected upper bound fraction. Center, median; box edges, 25th and 75th percentiles; whiskers, 5th and 95th percentiles. **f**, Fractions of UKB participants with the given numbers of genes affected by rare CNVs.

decrease in hypertension risk (OR = 0.83 (0.75–0.92), $P$ = 0.00026; Extended Data Fig. 2d and Supplementary Table 4a). The strongest BP association at this locus was attained by a common *RGL3* missense variant (rs167479; AF = 47%) independent of the deletion ($r^2$ = 0.005; Fig. 3a). Conditioning on rs167479 resulted in the deletion becoming the lead variant (Fig. 3a), supporting the causality of both *RGL3* coding variants and explaining a previously reported association of an intronic SNP in *RAB3D* (rs55670943, 76 kb downstream[32]) that best tags the deletion ($r^2$ = 0.66).

The deletion variant had a much larger effect on BP than the missense variant, similar to the effect of a rare *RGL3* stop gain (Fig. 3b), suggesting that it causes loss of RGL3 function. Analysis of RNA sequencing (RNA-seq) data from the Genotype-Tissue Expression (GTEx) project[33] provided insight into the transcriptional basis for this effect: carriers of the deletion, which removes the exon 6 splice acceptor, exhibited splicing into a new splice acceptor upstream of the deletion (Fig. 3c), translating to an inframe substitution of a new 23-amino acid sequence for a 19-amino acid segment of RGL3. Further work will be required to determine whether the modified protein is completely dysfunctional or whether the apparent LOF effect is mediated in part by reduced expression of *RGL3* alleles carrying the deletion (Supplementary Note).

Intriguingly, the BP-lowering effect of the deletion in *RGL3* is one of the strongest such effects among all coding variants genome-wide (Fig. 3d); knockout of *RGL3* is likely well tolerated based on the presence of 37 UKB participants homozygous for the deletion who appeared to be generally healthy (Supplementary Note). These observations raise the possibility that RGL3, or a pathway in which it functions, could be an appealing target for antihypertensive drug development, motivating further study of RGL3 function.

### Identifying the impacts of common coding copy number variation

In addition to the genetic effects above, in uniquely mappable regions of the human genome, potentially important effects on human biology could arise within rapidly evolving gene families shaped by extensive recent gene duplication and divergence. The analytical technique above was designed to detect rare protein-altering CNVs within mappable regions. To enable exploration of common coding copy number variation, including abundant variation within segmental duplications[34], we developed another approach that first identifies genomic regions that harbor common copy number variation (based on correlated WES read depth among parent–child trios)

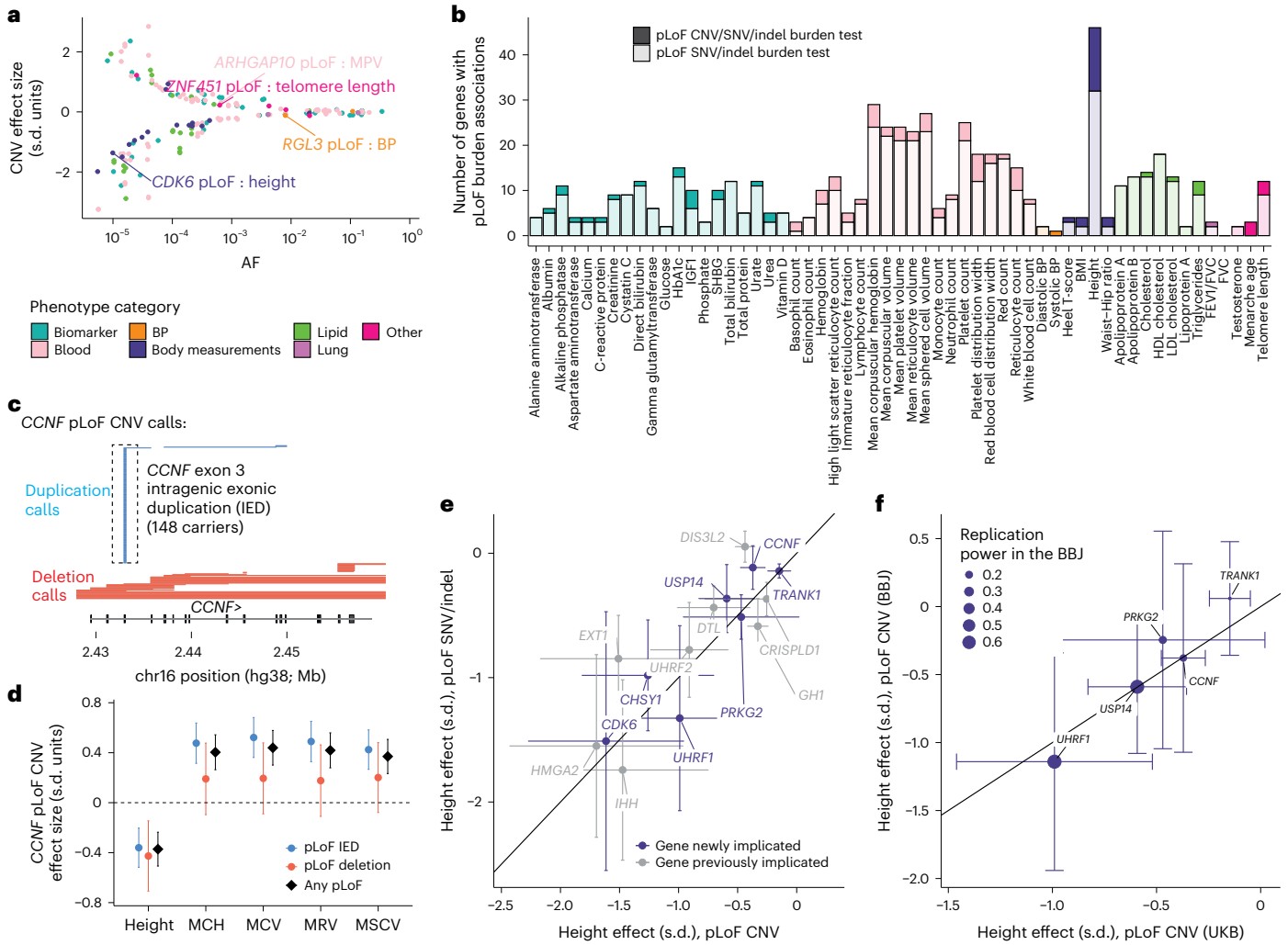

**Fig. 2 | Association and fine-mapping analyses implicate rare large-effect CNVs and uncover new gene–trait relationships. a**, Effect size versus minor allele frequency (MAF) for 180 likely causal CNV–phenotype associations, colored according to phenotype category. **b**, Number of genes with pLOF burden associations ($P < 5 \times 10^{-8}$) per trait, colored according to phenotype category, with darker shading corresponding to associations detectable only when including pLOF CNVs (that is, $P > 5 \times 10^{-8}$ for burden masks considering only SNVs and indels). $P$ values are provided in Supplementary Data 3. **c**, Genomic locations of *CCNF* pLOF CNV calls; boxed calls correspond to the rare duplication

spanning a single 107-bp exon. **d**, Effect sizes of *CCNF* pLOF CNVs for height and erythrocyte traits. MCH, mean corpuscular hemoglobin; MCV, mean corpuscular volume; MRV, mean reticulocyte volume; MSCV, mean sphered cell volume. **e**, Consistency of height effect sizes of pLOF CNVs with those of pLOF SNVs and indels. **f**, Replication of height effect sizes of pLOF CNVs in the BBJ (for newly implicated genes with at least five pLOF CNV carriers in the BBJ). The error bars represent the 95% confidence intervals (CIs). Sample sizes for the UKB (**d**–**f**) are reported in Supplementary Data 1; $n = 179,420$ for the BBJ.

and then measures copy number in these regions by leveraging haplotype sharing to denoise read depth-derived estimates. This approach generalizes the techniques we recently developed to study variable number tandem repeat (VNTR) polymorphisms[21]; here, we developed new algorithms to analyze a much larger set of CNV regions (Methods).

This approach detected 41,042 genomic regions (defined at the resolution of 100-bp segments, exons or previously reported CNVs) with evidence of common copy number-altering structural variation. These commonly copy number-variable regions overlapped coding exons of 11% of autosomal genes, which tended to have lower probability of LOF intolerance (average pLOF intolerance = 0.16 across such genes versus 0.25 across genes not impacted by common copy-altering SVs; Supplementary Table 5).

Measuring copy number variation in these regions, many of which are invisible to large-scale genetic analysis pipelines, provided a unique opportunity to search for associations with phenotypic variation in the UKB. Given the difficulty of modeling potentially complex structural

variation in such regions, compounded with the challenge of analyzing short-read alignments in low-mappability regions, we performed association analyses on quantitative, dosage-like measurements derived from read depth rather than attempting to call discrete genotypes (Extended Data Fig. 1). We reasoned that while these measurements might only roughly represent SV alleles, association signals could still point to phenotypically important SV regions meriting more careful follow-up.

This strategy proved fruitful: association analyses with 57 quantitative traits identified 375 associations at 99 loci not explainable by linkage disequilibrium (LD) to nearby SNPs (Supplementary Data 4), recovering strong VNTR–phenotype associations we recently reported (including a 39-bp coding repeat in *GP1BA* associated with platelet traits[35]; $P = 1.1 \times 10^{-133}$), and revealing several new loci involving multicopy variation poorly tagged by SNPs. Follow-up analyses of the most intriguing associations, detailed below, enabled further exploration of genetic variation at these loci and its influences on human health.

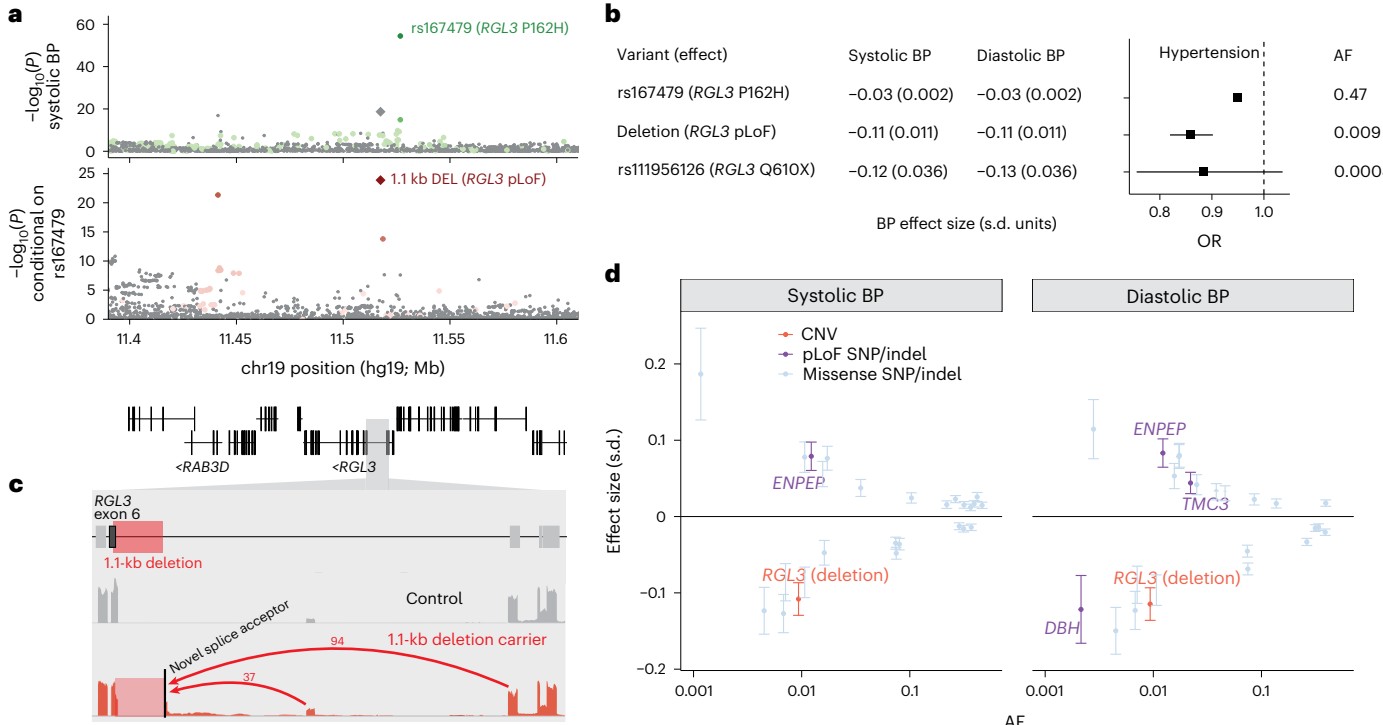

**Fig. 3 | A low-frequency deletion in *RGL3* is associated with reduced hypertension risk and alters splicing. a**, Associations of variants at the *RGL3* locus with systolic BP in two steps of stepwise conditional analysis. The colored dots are variants in partial LD ($r^2 \geq 0.01$) with labeled variants. **b**, Effect sizes and AFs of a common *RGL3* missense variant (rs167479), the low-frequency 1.1-kb deletion and a rare *RGL3* stop gain. **c**, Evidence of aberrant *RGL3* splicing produced by the 1.1-kb deletion. RNA-seq read-depth data from the GTEx are shown for a carrier of the deletion and a control sample (both thyroid); the red arcs indicate new splice junctions, labeled with counts of supporting RNA-seq reads. **d**, Systolic and diastolic BP effect sizes versus MAFs for nonsynonymous SNP and indel variants and the 1.1-kb deletion. The error bars represent the 95% CIs. Sample sizes for BP are reported in Supplementary Data 1; *n* = 437,475 for hypertension.

## Common variants in segmental duplications modulate type 2 diabetes risk

Copy number variation at 7q22.1 and in *CTRB2* associated with HbA1c and type 2 diabetes (T2D), contributing two of the top 20 T2D-associated loci in UKB (Fig. 4a). The 7q22.1 locus, which also generated the human genome's strongest association with chronotype (Fig. 4b), contains a 99-kb segmental duplication that is among the largest, most polymorphic CNVs in the human genome[5] and encompasses four protein-coding genes (Fig. 4c). While copy number of the segment (typically ranging from two to 14 copies per diploid genome) associated with T2D status ($P = 2.4 \times 10^{-13}$ in the UKB; Fig. 4c,d; replication $P = 2.0 \times 10^{-4}$ in AoU; Supplementary Table 4b), we wondered whether this signal might be driven by paralogous sequence variants (PSVs), that is, SNPs and indels carried on one or more copies of the 99-kb segment within each allele. To genotype such variation, which is inaccessible to conventional analysis of short-read data, we first roughly estimated PSV genotypes from WGS read alignments (available for 200,018 UKB participants[18]) and then adapted our haplotype-informed approach to denoise PSV genotypes and impute them into the remainder of the UKB cohort (Extended Data Fig. 4 and Methods).

Intriguingly, testing PSVs at 7q22.1 for association with T2D and chronotype identified a common missense PSV in *RASA4* (encoding Ras GTPase-activating protein 4) as the most strongly associated variant for T2D and second strongest for chronotype ($P = 1.3 \times 10^{-25}$ and $2.6 \times 10^{-72}$, respectively; Supplementary Table 6a; T2D replication $P = 2.8 \times 10^{-5}$ in AoU; Supplementary Table 4b). For both phenotypes, the number of copies of *RASA4* with this variant (encoding a Y731C substitution in the canonical transcript) associated much more strongly than copy number of the 99-kb segment (Fig. 4c); for chronotype, the *RASA4* missense PSV associated far more strongly than variants at all other loci across

the genome (Fig. 4b). The contribution of this locus to each phenotype had largely been hidden from previous analyses because SNPs flanking the segmental duplication poorly tag the multicopy variation within it (Fig. 4c). The total number of copies of *RASA4* carrying the Y731C missense PSV (typically ranging from 0 to 3 per individual; Fig. 4d) was associated with increasing T2D risk and 'eveningness' (that is, later preferred bedtime and rising time) (Fig. 4e), with a 1.30-fold (1.21–1.39) range in odds of T2D. This PSV is a strong candidate causal variant given its protein-altering effect and support from statistical fine-mapping (Supplementary Note); however, further study is required to determine whether it indeed underlies one or both of these associations, and if so, how this variant affects *RASA4* function.

The *CTRB2* gene encodes the chymotrypsinogen B2 protein, which is primarily produced in the pancreas, is converted into the active enzyme chymotrypsin B through enzymatic cleavage in the small intestine and has an important role in the digestive process[36]. A common 584-bp deletion (AF = 0.08) spanning exon 6 of *CTRB2* underlies another top locus for T2D (Fig. 4a,f). This deletion falls within a region of high homology to *CTRB1*, but our analysis pipeline successfully captured the copy number variability of exon 6 from WES read depth despite the low mappability (Fig. 4g,h). The deletion associated with decreased T2D risk ($P = 1.6 \times 10^{-16}$, strongest at the locus; OR = 0.86 (0.82–0.89)), replicating in AoU ($P = 2.3 \times 10^{-5}$; Extended Data Fig. 2e and Supplementary Table 4c). We also replicated a recently reported association of the deletion (which was shown to impair chymotrypsin B2 function and localization) with increased risk of pancreatic cancer[37] ($P = 4.2 \times 10^{-12}$; Fig. 4i and Supplementary Table 6b). The opposite effect direction of these associations is notable given the overall epidemiological association of T2D with increased pancreatic cancer risk[38].

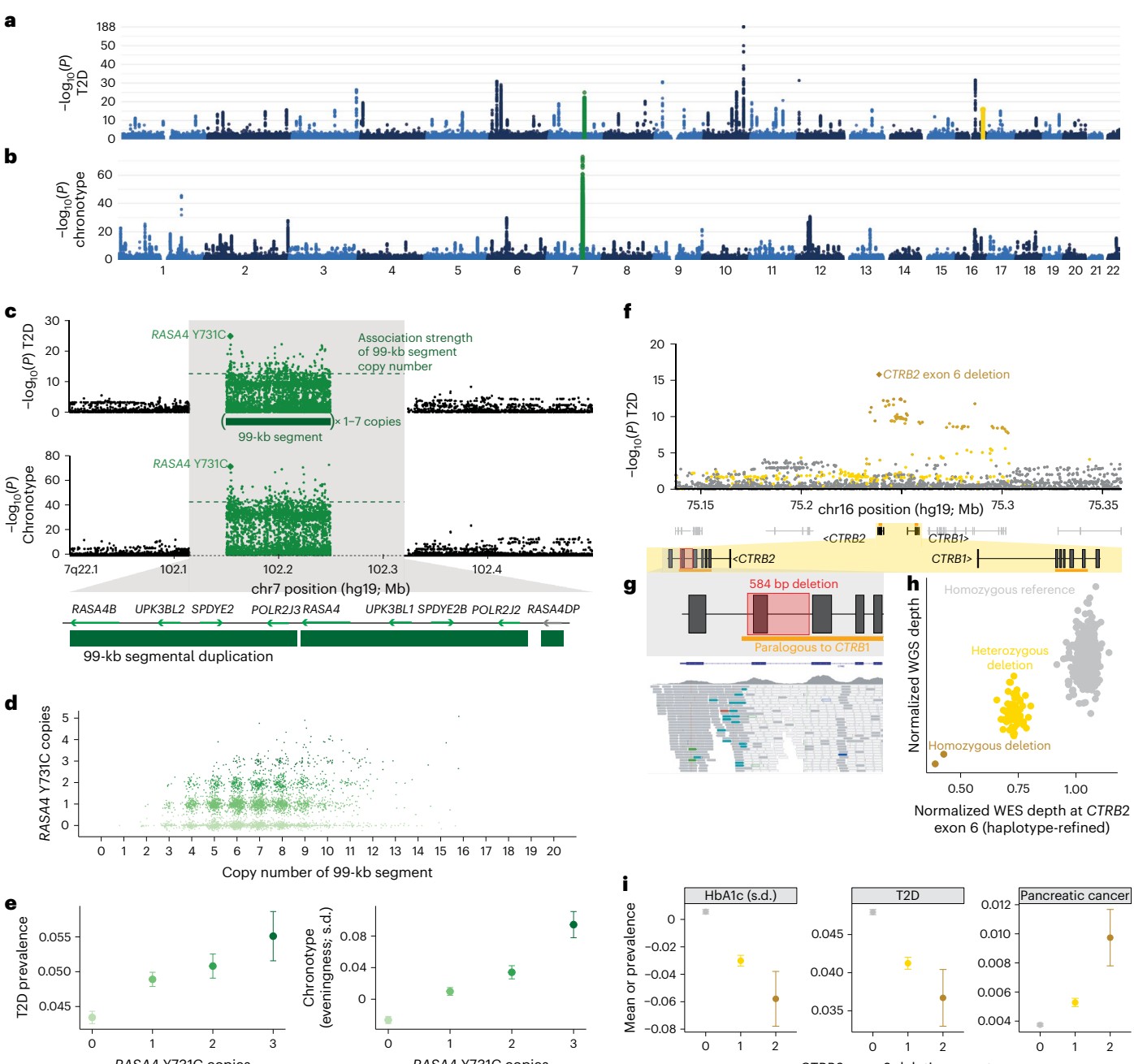

**Fig. 4 | Coding variants within segmental duplications underlie top genetic associations with T2D and chronotype. a,b**, Genome-wide associations with T2D (**a**) and chronotype (**b**). **c**, Associations of variation at 7q22.1 with chronotype and T2D. Associations of PSVs within the 99-kb repeat at this locus (1–7 copies per allele; two copies in GRCh37) are plotted in the center; the green dashed line indicates the association strength of copy number of the 99-kb repeat. **d**, Joint distribution of copy number estimates for the 99-kb segmental duplication and the *RASA4* Y731C missense variant. **e**, T2D prevalence and mean chronotype (in standardized units; higher for 'evening people') as a function of the number of copies of the *RASA4* Y731C missense variant. **f**, T2D associations

at the *CTRB2* locus; the colored dots are variants in partial LD ($r^2 > 0.01$) with the *CTRB2* exon 6 deletion. **g**, Location of the 584-bp deletion spanning *CTRB2* exon 6 (top) and exome sequencing read alignments for a deletion carrier (bottom); most reads aligned to the region paralogous to *CTRB1* do not map uniquely and are colored white. **h**, Scatter plot of normalized whole-genome and WES read depths at *CTRB2* exon 6. **i**, Mean HbA1c and prevalence of T2D and pancreatic cancer as a function of *CTRB2* exon 6 deletion genotype. The error bars represent the 95% CIs. The sample sizes for HbA1c are reported in Supplementary Data 1; $n = 453,585$ for T2D; $n = 454,633$ for pancreatic cancer; $n = 406,359$ for chronotype.

## Segmental duplication variants associate strongly with blood traits

Copy number variation in two other segmental duplication regions produced two of the top five independent associations with count of basophils (Fig. 5a), a type of white blood cell that has a role in the immune response and the regulation of allergic reactions. In this study, our analysis helped to recognize powerful effects within the

*FCGR3* gene family, which encodes a family of cell surface receptors found on several immune cells, including neutrophils, macrophages and natural killer cells; FCGRs have a crucial role in the immune response by recognizing and binding to the Fc portion of immuno-globulins (antibodies) that are bound to antigens[39]. In the UKB, the copy number of *FCGR3B* (which our analysis disambiguated from that of its paralog, *FCGR3A*) associated strongly with increased basophil count

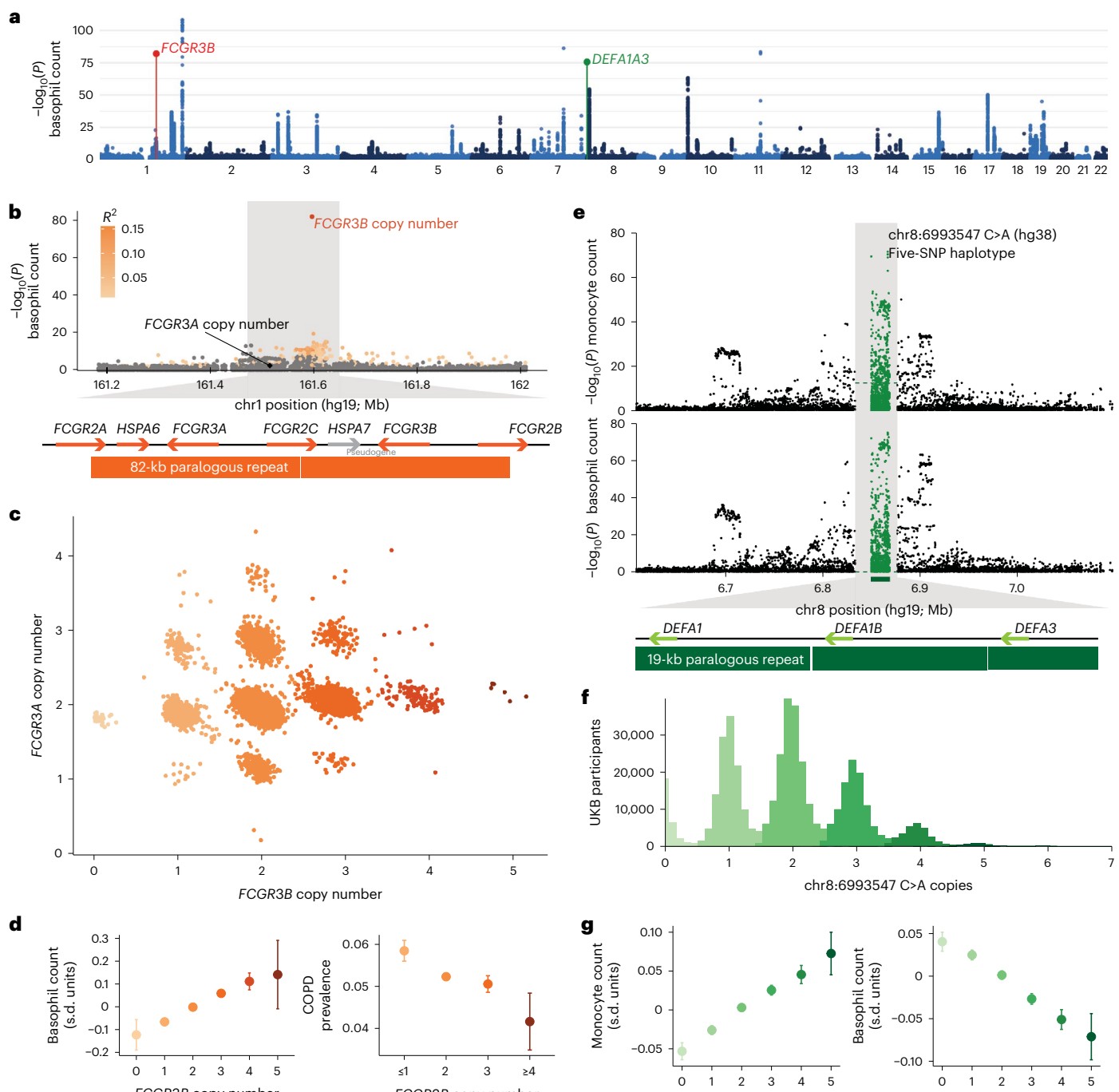

**Fig. 5 | Variation in segmental duplications generates two of the top five genetic associations with basophil counts. a**, Genome-wide associations with basophil counts. **b**, Associations with basophil counts at the *FCGR3B* locus; the colored dots are variants in partial LD ($r^2 > 0.01$) with the *FCGR3B* copy number. **c**, Joint distribution of copy number estimates for *FCGR3A* and *FCGR3B*. **d**, Mean basophil count and prevalence of chronic obstructive pulmonary disease (COPD) as a function of *FCGR3B* copy number. **e**, Associations with basophil counts at the

*DEFA1A3* locus. PSVs within the 19-kb repeat at this locus are plotted as in Fig. 4c; the green dashed line indicates the association strength of the copy number of the 19-kb repeat. **f**, Histogram of the number of copies of the 19-kb repeat carrying the five-SNP haplotype represented by chr8:6993547 C>A (GRCh38 coordinates). **g**, Mean monocyte and basophil count as a function of copy number of the five-SNP haplotype. The error bars represent the 95% CIs. Sample sizes for blood counts are reported in Supplementary Data 1; $n = 454,633$ for COPD.

($P = 1.4 \times 10^{-82}$, far exceeding the associations of nearby SNPs, which poorly tagged the recurrent CNV; Fig. 5b,c). Analysis of FCGR3B plasma protein levels corroborated the *FCGR3B* genotypes (Extended Data Fig. 5). *FCGR3B* deletion has previously been associated with several autoimmune disorders[39–41]; in this study, decreasing *FCGR3B* gene dosage was also associated with an increasing risk of chronic obstructive pulmonary disease ($P = 7.5 \times 10^{-7}$; Fig. 5d and Supplementary Table 6c).

The *FCGR* locus on 1q23.3 contains many functional variants, including multiple distinct CNVs[39], such that while the basophil count association is driven by the *FCGR3B* copy number, other associations at this locus (Supplementary Data 4) may reflect other causal variants.

We also recognized potent effects within the family of alpha-defensin genes, a rapidly evolving gene family that encodes a class of small, cationic peptides that are part of the innate immune

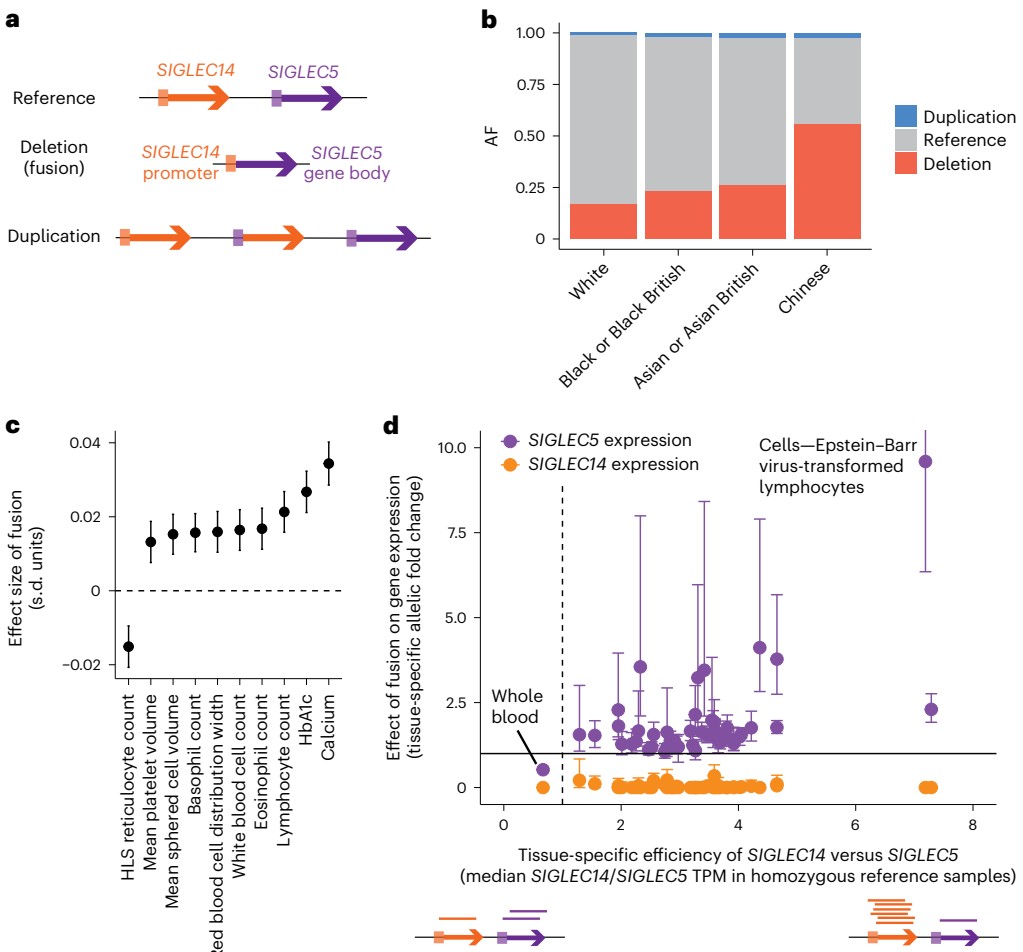

**Fig. 6 | Common pleiotropic *SIGLEC14*–*SIGLEC5* gene fusion illustrates tissue-specific promoter activity. a**, Gene diagram of *SIGLEC14* and *SIGLEC5*. A common deletion allele fuses the *SIGLEC14* promoter to the *SIGLEC5* gene body, and the reciprocal duplication allele is also observed at lower frequencies. **b**, Allele frequency of gene fusion and duplication events in the UKB, stratified according to reported ethnicity. **c**, Effect size of fusion on blood indices and serum biomarker traits. **d**, Allelic fold change effect of fusion on *SIGLEC5* and *SIGLEC14* gene expression across GTEx tissues tracked with relative efficiency the *SIGLEC14* versus *SIGLEC5* promoter in each tissue. TPM, transcript per million reads. The error bars represent the 95% CIs. Sample sizes for the blood indices and serum biomarker traits are reported in Supplementary Data 1; sample sizes for the allelic fold changes are reported in Supplementary Table 7.

system and have a crucial role in host defense against microbial infections[42]. Variation at the alpha-defensin gene cluster at 8p23.1 associated strongly with basophil count (Fig. 5a,e) as well as monocyte count (Fig. 5e). Alleles at this locus contain a highly variable number of copies of a 19-kb repeat, each containing a single alpha-defensin gene[43]. Analysis of PSVs within this region (which had not previously been studied at scale, similar to the *RASA4* locus at 7q22.1) suggested that the number of copies of the 19-kb segment carrying a tightly linked five-SNP haplotype within an Alu element inside the repeat, rather than the total number of copies of the repeat, might drive the association (Fig. 5e and Supplementary Table 6d). The number of copies of this repeat type typically ranged from 0 to 5 per individual (Fig. 5f) and associated with steadily increasing monocyte count and decreasing basophil count (Fig. 5g); however, we caution that a functional consequence of the five-SNP haplotype is not immediately clear, unlike for the protein-coding variants at other loci we have highlighted.

### *SIGLEC14*–*SIGLEC5* gene fusion demonstrates tissue-specific promoter activity

A common, pleiotropic CNV at the *SIGLEC14*–*SIGLEC5* locus provided a unique opportunity to isolate a tissue-specific effect of a promoter element. A deletion allele at this locus that is particularly common in East Asians creates a fusion gene in which *SIGLEC5* is placed under the

control of the *SIGLEC14* promoter[44] (Fig. 6a,b). In the UKB, this CNV is associated with several blood cell indices and serum biomarkers ($P = 1.5 \times 10^{-8}$ to $1.7 \times 10^{-37}$; Fig. 6c, Supplementary Data 4 and Supplementary Table 7a). Follow-up analysis in the GTEx dataset revealed an unusually tissue-specific effect of the fusion on *SIGLEC5* expression, with the effect size varying greatly in magnitude and even direction across tissues (Fig. 6d). This phenomenon was explained by the further observation that the fusion's tissue-specific effects on *SIGLEC5* expression tracked with relative efficiency the *SIGLEC14* and *SIGLEC5* promoters across tissues (measured by the relative expression of *SIGLEC14* and *SIGLEC5* in individuals homozygous for the reference allele), such that the variable effect of the fusion was in fact consistent with its substitution of the *SIGLEC14* promoter in place of the *SIGLEC5* promoter (Fig. 6d and Supplementary Table 7b).

Other notable results included two strong associations with leukocyte telomere length, one involving an 84-bp deletion within an alternative last exon of *ZNF208* ($P = 1.7 \times 10^{-53}$) and the other involving difficult-to-resolve copy number variation in the *CLEC18A/CLEC18B/CLEC18C* gene family ($P = 1.0 \times 10^{-40}$), which exhibits complex structural variation across two loci more than 4 Mb apart[16]. Future analyses of long-read datasets will be better able to probe variation at such segmental duplications and elucidate phenotypic consequences hinted at in this study.

## Discussion

These results illustrate the phenotypic impact of protein-altering CNVs hidden from large-scale analyses to date. In this study, we observed that such variants include top genetic influences on human phenotypes that have eluded genetic association studies despite steadily increasing sample sizes and phenotyping precision. We further identified new gene–trait relationships implicated by rare CNVs that, for many genes, account for a substantial proportion of LOF events. We caution that some of these associations still need replication; in this study, we replicated a subset of the associations and observed corroborating evidence from allelic series for others. Additionally, while the protein-coding variants we have implicated have clear effects on amino acid sequence or gene dosage, experimental work is needed to confirm causality of these variants and understand how they influence function and ultimately phenotype.

Our analyses, based on exome sequencing of 468,570 individuals in the UKB, are far from comprehensive. While our haplotype-informed approach accurately recognized several subexonic CNVs that we linked to phenotypes, we expect that it missed very rare short CNVs carried by only a few UKB participants. We also did not attempt to study shorter tandem repeats, which require specialized techniques[45]. Additionally, our analysis of common CNV regions—via rough quantifications of copy number—imperfectly modeled complex, multiallelic structural variation. More precise genotyping of variation in such regions is needed, particularly in segmental duplications (approximately 7% of the human genome[46]). Our analyses were also limited in scope by the generally healthy, predominantly European ancestry composition of the UKB cohort. A search for associations between disease traits and gene-inactivating variants (including CNVs) only recovered known Mendelian disease genes (Supplementary Data 5), reflecting the limited power to study rare diseases in population cohorts. Finally, while we prioritized compelling associations to highlight in this article using a stringent statistical fine-mapping filter, relaxing this filter would yield many more associations.

We anticipate that expanding genome sequencing projects, including some that will use long reads[17,47], will overcome many of these limitations, and we look forward to further insights into phenotypic consequences of both coding and noncoding structural polymorphisms in the years ahead.

## Online content

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

## Methods

### Ethics

This research complies with all relevant ethical regulations. The study protocol was determined to be not human subjects research by the Broad Institute Office of Research Subject Protection and the Partners HealthCare Human Research Committee as all data analyzed were previously collected and de-identified.

### UKB genetic data

WES data were previously generated for approximately 470,000 UKB participants[23]. We analyzed these data together with SNP haplotypes we previously generated for 487,409 participants[48] in the UKB SNP array and imputation data set (imp_v3)[22]. We performed haplotype-informed CNV detection on all UKB participants with SNP haplotypes, including 468,570 participants with WES data passing quality control and the remaining 3% of the imp_v3 samples using an imputation approach (Supplementary Note). We also analyzed WGS data available for 200,018 participants[18] in validation and follow-up analyses of PSVs within segmental duplications. We focused our primary analyses on individuals of self-reported White ethnicity (previously shown to be predominantly of European genetic ancestry[22]), excluding individuals with trisomy 21, blood cancer, with aberrantly many CNV calls and those who had withdrawn at the time of our study (Supplementary Note), resulting in 454,682 participants being included in main analyses.

### UKB phenotype data

We primarily analyzed 57 heritable quantitative traits measured on most UKB participants (Supplementary Data 1), including 56 quantitative traits we recently analyzed[9] along with telomere length. We reprocessed blood traits using a slightly modified pipeline in which we did not perform outlier removal because some rare variants produce extreme blood indices: that is, within strata of sex and menopause status, we performed inverse normal transformation and then regressed out age, ethnicity, alcohol use, smoking status, height and body mass index[9]. We processed the telomere length phenotype (Data-Field 22192)[49] by applying inverse normal transformation. The remaining traits were processed as described previously[9].

In secondary analyses (for example, follow-up at the loci of interest), we analyzed additional traits including binary disease outcomes derived from self-report (touchscreen questionnaire at assessment), hospital inpatient records, and cancer and death registries, as well as plasma protein abundance for FCGR3B. In particular, we analyzed hypertension (174,773 cases and 279,891 controls; first-occurrence Data-Field 131286), T2D (21,292 cases and 432,324 controls; derived from self-reported, doctor-diagnosed T2D Data-Field 2443, according to ref. [50]), pancreatic cancer (1,816 cases and 452,848 controls; International Statistical Classification of Diseases and Related Health Problems, 10th revision code C25 from hospital records and cancer and death registries) and COPD (23,875 cases and 430,789 controls; first-occurrence Data-Field 131492). Further details are provided in the Supplementary Note.

### Replication datasets

We replicated key genetic associations in the BBJ[31] and AoU[47] cohorts. For rare pLOF CNV associations with height, we performed replication analyses in the BBJ ($n = 179,420$) using a SNP array-based CNV call set we generated previously[9]. For associations with hypertension (at *RGL3*) and T2D (at *RASA4* and *CTRB2*), we performed replication in AoU by genotyping each variant under consideration from high-coverage WGS data ($n = 245,394$ in the AoU v7 release). Additionally, for variants with potential transcriptional effects (at *RGL3* and *SIGLEC14/SIGLEC5*), we performed follow-up in the GTEx[33] dataset ($n = 838$ in GTEx v8). Details are provided in the Supplementary Note.

### Overview of hidden Markov model method for haplotype-informed rare CNV detection

CNV calling from exome sequencing data typically involves searching for consistent increases or consistent decreases in a sample's WES read coverage across a series of captured genomic regions, indicating the presence of a duplication or a deletion. This requires accurately modeling WES read coverage, which can be substantially influenced by technical differences in exome capture that may vary across samples and across genomic regions (for example, because of heterogenous effects of local GC content). While exome sequencing of the UKB was performed relatively uniformly across samples, exome capture was performed using a different Integrated DNA Technologies oligonucleotide lot for the first 50,000 samples[51] versus the remainder of the cohort, requiring careful treatment of this batch covariate.

Our overall strategy to account for technical variation in WES read coverage (both across and within oligonucleotide lots) was to estimate sample-specific baseline models of expected read depth by identifying sets of reference samples with best-matching exome-wide coverage profiles[21]. We analyzed WES read coverage at the resolution of 100-bp bins, restricting to bins with coverage in both oligonucleotide lots, similar coverage across the two oligonucleotide lots and sufficient mappability (requiring most aligned reads to have positive mapping quality). To optimize for robust analysis of rare CNVs, we further restricted to bins in which we could accurately calibrate normalized read coverage to absolute copy number, either because a bin was rarely affected by copy number variation or because discrete copy number states could be confidently identified.

While most WES-based CNV callers analyze each sample independently after performing normalization, we reasoned that we could increase CNV detection sensitivity by integrating WES data across individuals likely to have co-inherited a large genomic tract (as in our recent SNP array-based CNV analysis[9]). Similar to our previous work, we used a hidden Markov model to call CNVs in this haplotype-informed way, integrating information regarding copy number state across an individual and up to ten 'haplotype neighbors' with expected time to most recent common ancestor less than a selected value (equivalently, if the length of identity by descent (IBD) sharing exceeded a threshold).

In more detail, for each 100-bp bin, for the individual and each haplotype neighbor, we used negative binomial distributions with sample-specific and region-specific parameters to estimate a Bayes factor for deletion and duplication states based on counts of read alignments within the 100-bp bin for each sample. For a given threshold on the minimum length of IBD sharing, we computed a haplotype-informed combined Bayes factor by multiplying Bayes factors across the target individual and all haplotype neighbors with IBD sharing passing the threshold. We ran this analysis using a set of different IBD length thresholds (trading off sensitivity to more recent versus older mutations) and compiled calls made across these IBD parameter values.

To obtain a high-quality CNV call set, we performed subsequent filtering of various classes of calls that tended to be of lower quality based on inspection of WES and WGS read alignments at initial calls in a pilot analysis. We also removed individuals with more than 300 CNV calls. For downstream association analyses, we masked calls on any chromosome in which we had previously called a mosaic CNV[48]. Further methodological details are available in the Supplementary Note.

### Validation of HMM-based CNV call set

To benchmark the precision of the HMM-based CNV call set, we analyzed independent WGS data for 100 individuals. For each of these individuals, we assessed whether or not WGS read depth was higher (respectively, lower) than expected within the putative duplications (respectively, deletions) called. We estimated the validation rate as the difference between the fraction of calls with WGS read depth in the correct direction versus the opposite direction, reasoning that false

positive calls should be equally likely to have WGS read depth in either direction by chance. We also determined precision and recall for the *CCNF* exon 3 duplication and *RGL3* exon 6 partial deletion by directly genotyping these CNVs using discordant read and breakpoint-based analyses, respectively (Supplementary Note).

## Overview of haplotype-informed analysis of common copy-altering SVs

The HMM pipeline above was designed primarily to robustly call rare CNVs from exome sequencing data. HMM approaches for this task directly model the read coverage generated from discrete copy number states, which can aid statistical power and breakpoint precision when the model is accurate. However, such approaches can produce suboptimal performance when model assumptions are violated. Model violations are especially prone to occur at common CNV loci (where calibration of read depth to copy number states can be challenging) and in segmental duplications (where high sequence homology can influence read mapping in ways that cause copy number alterations to have unexpected effects on read depth, and loci may contain multiple complex SVs). For these reasons, genotyping common CNVs from short-read sequencing, especially within segmental duplications, is technically challenging and requires careful modeling[52], such that general-purpose SV analysis pipelines deployed at scale have had limited ability to assess such variation[3,24].

Despite these challenges, short-read sequencing read depth, including from WES, contains useful signatures of common CNVs and other copy-altering SVs, such as VNTRs. We reasoned that, even if precisely characterizing such variants from WES is intractable, the signals of copy number variation contained in WES read-depth data could still provide approximations of structural variation that, while rough, could enable discovery of CNV loci associated with phenotypes, after which the variants involved could be precisely resolved through follow-up analyses of WGS or long-read data. Therefore, we developed a complementary analysis pipeline to roughly estimate copy number variation across individuals (measured on a continuous rather than discrete scale) from WES read coverage at a broad set of predefined genomic regions (including 100-bp bins, exons and previously reported CNVs), extending methods we recently developed to analyze VNTRs[21].

For each genomic region under consideration, we counted the WES reads aligned to the segment and normalized these read counts using sample-specific reference panels with matched exome-wide coverage profiles (as in the first step of the HMM pipeline). Unlike the HMM pipeline, we considered low-mappability regions (for example, within segmental duplications) and we generated two read count measurements per region, one counting all reads regardless of mapping quality and the other counting only reads with positive mapping quality. We then evaluated which of these measurements appeared to be heritable, potentially reflecting common CNVs or other copy-altering SVs in the region. To do so, we computed mid-parent versus child correlations of normalized read counts in 704 trios, restricting further analysis to 100-bp bins and exons with significant correlation and all previously reported CNVs.

For each WES read count measurement that potentially represented common copy number variation in a region, we used long shared SNP haplotypes to statistically phase and simultaneously denoise the values measured across the UKB WES samples and also impute into individuals without WES data. To do so, we adapted the computational methods we previously used to analyze VNTRs[21], improving scalability by using the positional Burrows–Wheeler transform[53] to rapidly identify shared haplotypes. To catch instances in which exome capture bias rather than copy number variation was responsible for heritable variation in WES coverage (for example, short haplotypes containing several SNPs colocated within a few hundred base pairs that influence capture efficiency), we restricted to regions for which WES and WGS read-depth measurements exhibited consistent signal. Further details are available in the Supplementary Note.

## Overview of haplotype-informed analysis of PSVs

Beyond measuring copy number at polymorphic segmental duplications, our computational approach also enabled analysis of PSVs within such segments: that is, SNPs and indels present on varying numbers of copies of a repeated segment within a single allele. To do so, we first roughly estimated PSV genotypes from counts of WGS read alignments supporting each base (that is, 'pileups') and then adapted our haplotype-informed approach to denoise PSV genotypes and impute them into the remainder of the UKB cohort (Extended Data Fig. 4).

In more detail, for a repeat segment of interest, we first identified all regions in the GRCh38 reference sequence paralogous to the repeat segment and extracted all WGS reads aligning to these regions. We then realigned these reads to a new reference containing only one copy of the repeat segment plus a small buffer sequence containing the beginning of a second copy of the same repeat, facilitating harmonized ascertainment and genotyping of all common PSVs. Specifically, we estimated PSV allele fractions (that is, the fraction of repeat units containing a given PSV) from pileup counts, which we then converted to absolute estimates of PSV copy number (that is, the number of repeat units containing a given PSV) by scaling an individual's total repeat copy number by PSV allele fraction. We then used long shared SNP haplotypes to phase PSV copy number and impute into individuals without WGS data using the same approach we used to analyze common CNVs. Further details are provided in the Supplementary Note.

## Association testing and statistical fine-mapping

We performed phenotype association analyses on three classes of CNVs derived from the HMM-based CNV call set (defined based on (1) 100-bp bin overlap, (2) gene overlap and (3) gene pLOF burden), as well as the continuous-valued estimates of common copy number variation derived from heritable WES coverage (Extended Data Fig. 1).

We conducted association tests for our primary set of 57 quantitative traits using BOLT-LMM[25,26] with assessment center, genotyping array, WES release (50,000, 20,000, 454,000, 470,000, none), sex, age, age squared and 20 genetic principal components included as covariates. We fitted the linear mixed model on SNP array-genotyped autosomal variants with MAF > $10^{-4}$ and missingness lower than 0.1 and computed association test statistics for the copy number measurements defined above; a similar pipeline produced association test statistics for SNP and indel variants imputed by the UKB (the imp_v3 release[22]). We included all participants with nonmissing phenotypes in the quality controlled European ancestry call set described above. We removed associations potentially explainable by LD with imputed SNPs and indels within 3 Mb (refs. 9,54) (Supplementary Note).

The associations that survived this filtering represented copy-altering SVs—primarily CNVs but also some VNTRs—likely to causally influence phenotypes. We annotated CNVs on this list as syndromic based on a previously curated list of pathogenic CNVs[55]. For associations of particular interest that arose from the analysis of common CNVs, we undertook follow-up in the UKB WGS or Human Pangenome Reference Consortium long-read assemblies[5] to more precisely resolve CNVs, after which we refined CNV genotypes (using optimized analyses of UKB WES or WGS data) and undertook PSV analyses as necessary. Further details on filtering of associations and follow-up analyses at loci of interest are provided in the Supplementary Note.

All *P* values reported for statistical tests throughout this article are two-sided and uncorrected for multiple hypothesis testing as we corrected for multiple hypotheses tested by applying Bonferroni-adjusted significance thresholds.

## Reporting summary

Further information on research design is available in the Nature Portfolio Reporting Summary linked to this article.

## Data availability

Summary statistics for the CNV–phenotype association tests are available at https://data.broadinstitute.org/lohlab/UKB_WES_CNV_sumstats/ and have been deposited at Zenodo (https://doi.org/10.5281/zenodo.10529671)[56]. Access to the following data resources used in this study can be obtained by application: UKB (http://www.ukbiobank.ac.uk/), BioBank Japan (https://biobankjp.org/en/), AoU (https://allofus.nih.gov/) and GTEx (via the database of Genotypes and Phenotypes, https://www.ncbi.nlm.nih.gov/gap/, accession no. phs000424.v8.p2).

## Code availability

The custom code used to perform the haplotype-informed CNV analysis of the UKB WES data has been deposited at Zenodo (https://doi.org/10.5281/zenodo.10529671)[56]. The following open source software packages were also used: samtools v.1.11, mosdepth v.0.2.5, bedtools v.2.27.1, BLAT v.35, Burrows–Wheeler Aligner v.0.7.17, HTSbox r345, PLINK v.1.9, PLINK v.2.0, BOLT-LMM v.2.4.1, REGENIE v.2.2.4, SuSiE v.0.12.27 and R v.3.6.3.

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

## Acknowledgements

We thank Y. Momozawa at the RIKEN Center for Integrative Medical Sciences and members of the BBJ Project, headquartered at the University of Tokyo Institute of Medical Science, for supporting this project. This research was conducted using the UKB resource under application no. 40709. M.L.A.H. was supported by a US National Institutes of Health (NIH) Fellowship F32 HL160061. R.E.H. and S.A.M. were supported by US NIH grant no. R01 HG006855. M.A.S. was supported by a US NIH Fellowship F31 MH124393. N.K. was supported by US NIH training grant no. T32 HG002295. A.R.B. was supported by a US NIH fellowship F31 HL154537. R.E.M. was supported by US NIH grant no. K25 HL150334. C.T. was supported by Japan Agency for Medical Research and Development grant nos. JP21ek0109555, JP21tm0424220, JP21ck0106642, JP23ek0410114 and JP23tm0424225, and Japan Society for the Promotion of Science KAKENHI grant no. JP20H00462. P.-R.L. was supported by US NIH grant nos. DP2 ES030554, R56 HG012698 and R01 HG013110, a Burroughs Wellcome Fund Career Award at the Scientific Interfaces and the Next Generation Fund at the Broad Institute of MIT and Harvard. The funders had no role in study design, data collection and analysis, decision to publish or preparation of the manuscript. Computational analyses were performed on the O2 High Performance Compute Cluster, supported by the Research Computing Group at Harvard Medical School (http://rc.hms.harvard.edu), the UKB Research Analysis Platform and the *All of Us* Researcher Workbench. The *All of Us* Research Program is supported by the NIH, Office of the Director: Regional Medical Centers: 1 OT2 OD026549; 1 OT2 OD026554; 1 OT2 OD026557; 1 OT2 OD026556; 1 OT2 OD026550; 1 OT2 OD026552; 1 OT2 OD026553; 1 OT2 OD026548; 1 OT2 OD026551; 1 OT2 OD026555; IAA no. AOD 16037; Federally Qualified Health Centers: HHSN 263201600085U; Data and Research Center: 5 U2C OD023196; Biobank: 1 U24 OD023121; The Participant Center: U24 OD023176; Participant Technology Systems Center: 1 U24 OD023163; Communications and Engagement: 3 OT2 OD023205; 3 OT2 OD023206; and Community Partners: 1 OT2 OD025277; 3 OT2 OD025315; 1 OT2 OD025337; and 1 OT2 OD025276. In addition, the *All of Us* Research Program would not be possible without the partnership of its participants. The GTEx project was supported by the Common Fund of the Office of the Director of the NIH, and by the National Cancer Institute, National Human Genome Research Institute, National Heart, Lung, and Blood Institute, National Institute on Drug Abuse, National Institute of Mental Health and National Institute of Neurological Disorders and Stroke.

## Author contributions

M.L.A.H. and P.-R.L. performed the statistical analyses and wrote the manuscript. M.A.S. helped perform the initial statistical analyses. C.T. performed the validation analyses within the BBJ. R.E.H., M.A.S., N.K., A.R.B., R.E.M. and S.A.M. helped with the interpretation of the analyses. R.E.H., M.A.S., N.K., A.R.B., R.E.M., C.T. and S.A.M. provided feedback on the manuscript.

## Competing interests

The authors declare no competing interests.

## Additional information

**Extended data** is available for this paper at https://doi.org/10.1038/s41588-024-01684-z.

**Correspondence and requests for materials** should be addressed to Margaux L. A. Hujoel or Po-Ru Loh.

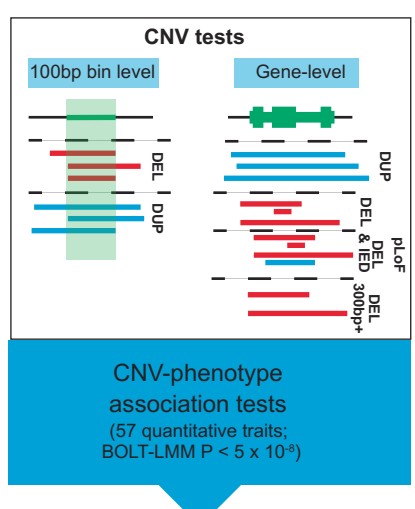

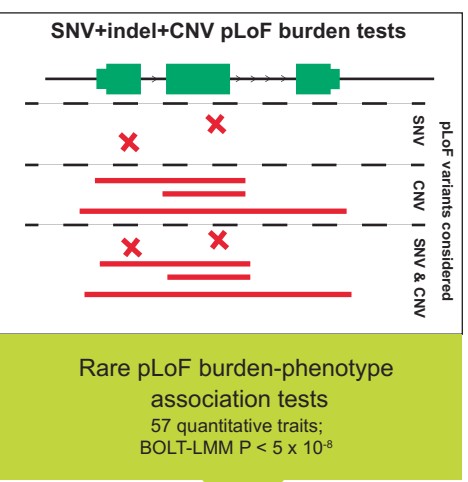

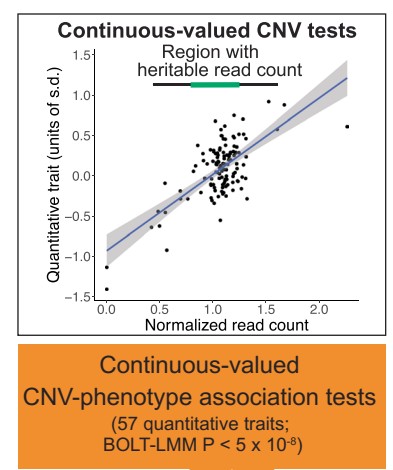

**Extended Data Fig. 1 | Overview of primary association analyses of 57 heritable quantitative traits.** Categories of variants and CNV measurements tested are depicted, and summary numbers of results from each set of association tests that remained after filtering are provided at the bottom.

(a) Read features to identify tandem duplications

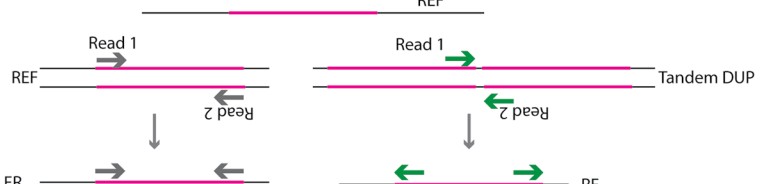

(b) Validation of *CCNF* exon 3 IED in UKB WGS

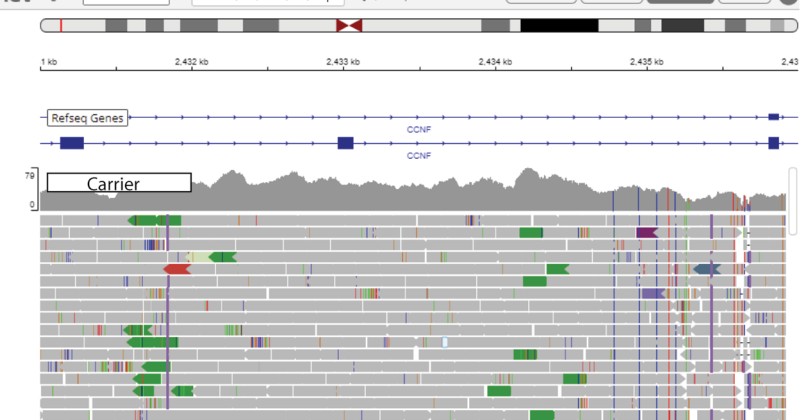

(d) *RGL3* genotyping in AoU

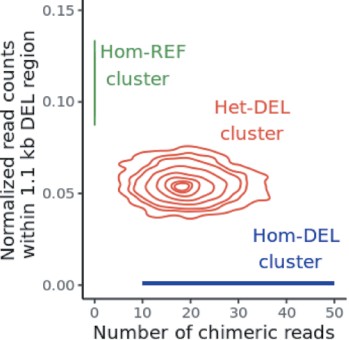

(e) *CTRB2* exon 6 DEL genotyping in AoU

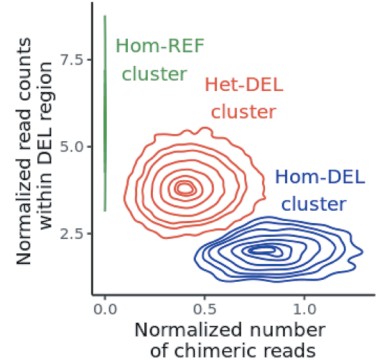

(c) Regenotype *RGL3* DEL to ascertain more likely DEL carriers in UKB (WES)

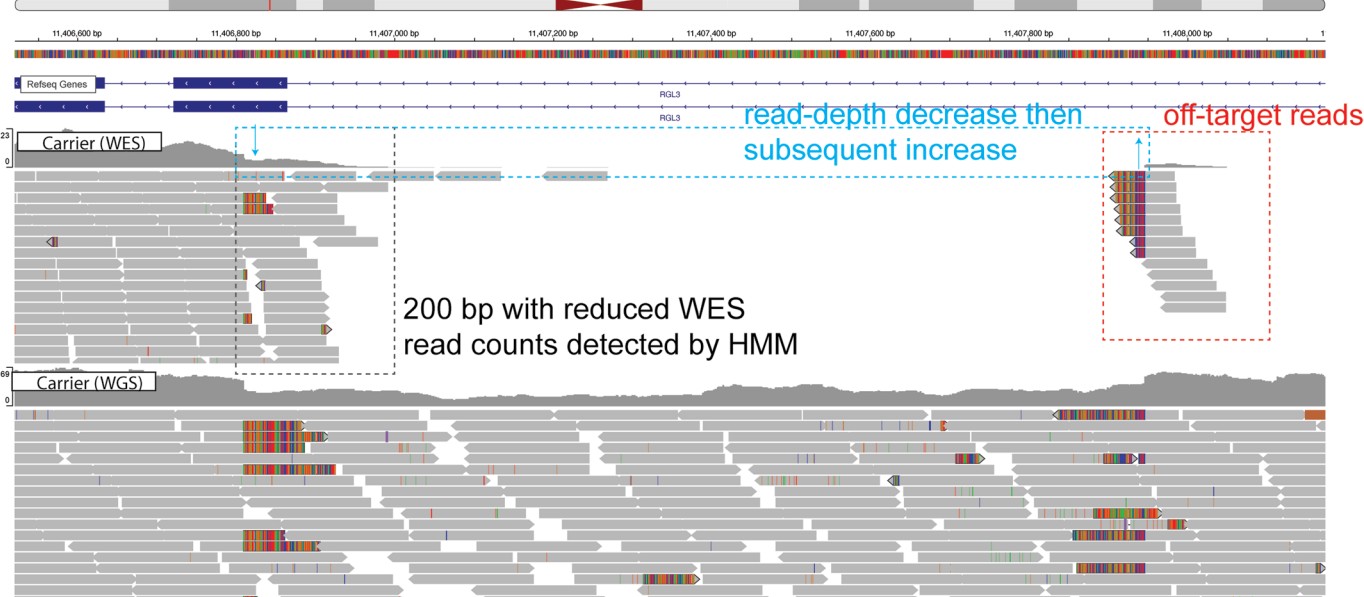

**Extended Data Fig. 2 | Additional CNV genotyping at key loci. a,** Schematic of discordant WGS reads that confirm tandem duplications and indicate breakpoint locations. **b,** We genotyped the *CCNF* exon 3 IED using discordant WGS reads (shown for a carrier in UKB) to assess precision and recall of WES-based calls from our HMM. **c,** IGV tracks of WES and WGS alignments for an *RGL3* deletion carrier. Top, WES features used in optimized breakpoint-based genotyping; bottom, independent confirmation of deletion from WGS. **d,e,** In *All of Us* (AoU), chimeric WGS reads and within-deletion read counts allowed the *RGL3* and *CTRB2* deletions to be cleanly genotyped. (The homozygous deletion cluster for *RGL3* contained <20 carriers, so to comply with AoU policy, the Hom-DEL line depicted is predicted from the heterozygous cluster).

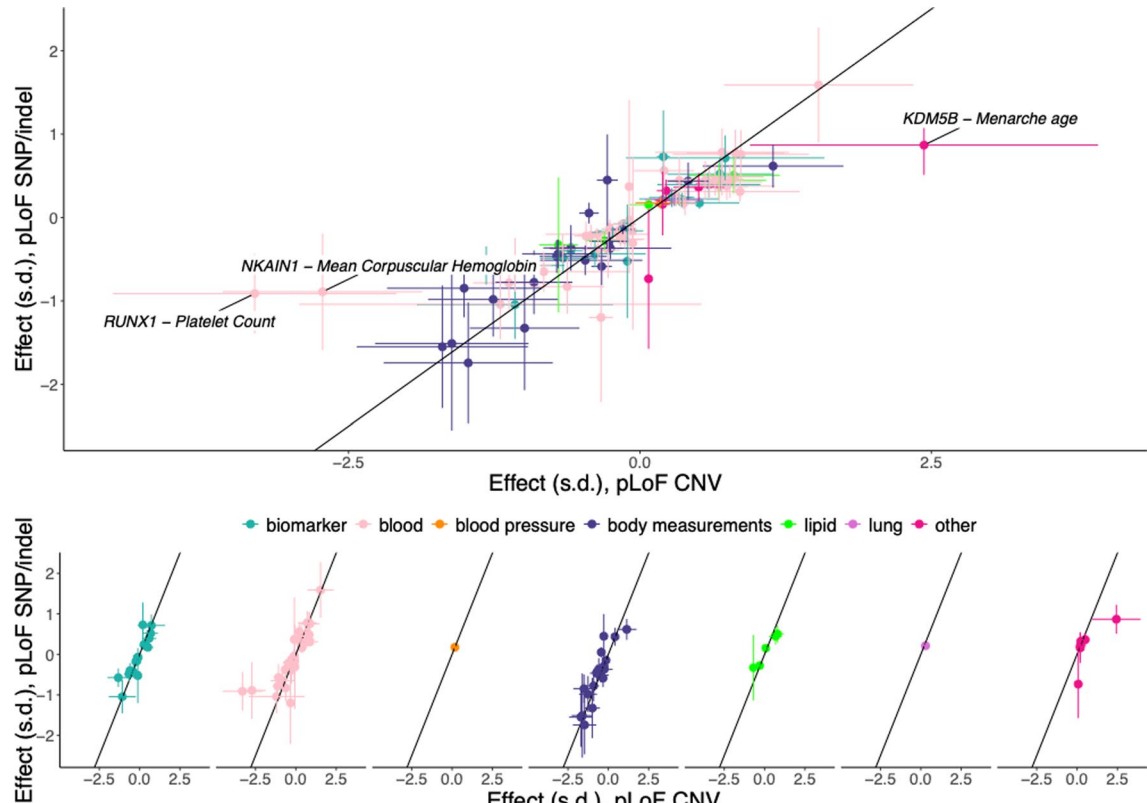

**Extended Data Fig. 3 | Consistency of effect sizes of pLoF CNV and SNP/indel variants across gene-trait associations.** Data are shown for all associations discovered only upon considering pLoF CNVs (i.e, not reaching significance in SNP/indel-only burden tests). The top plot is a merge across all traits, and the bottom plots show each phenotype category separately. Error bars are 95% confidence intervals. Sample sizes are reported in Supplementary Data 1.

# PSV analysis:

Regions with high sequence similarity (may also be copy number variable)

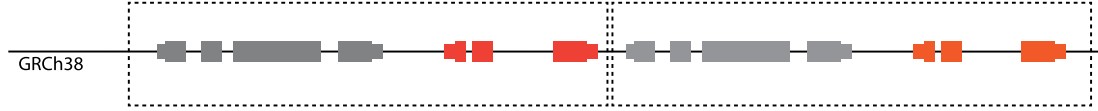

(1) Identify regions across the genome which have sequence similarity to our region of interest which may act as a "bait" for reads

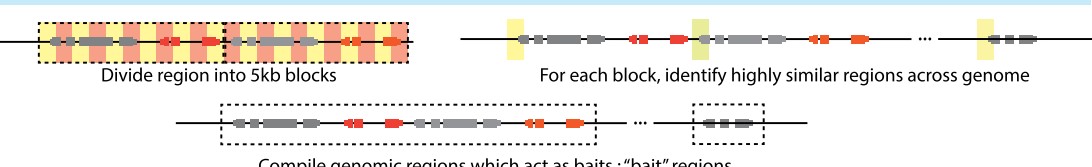

Divide region into 5kb blocks          For each block, identify highly similar regions across genome

Compile genomic regions which act as baits ; "bait" regions

(2) For all reads mapping to any of the "bait" regions; re-align to a reference consisting of one repeat unit

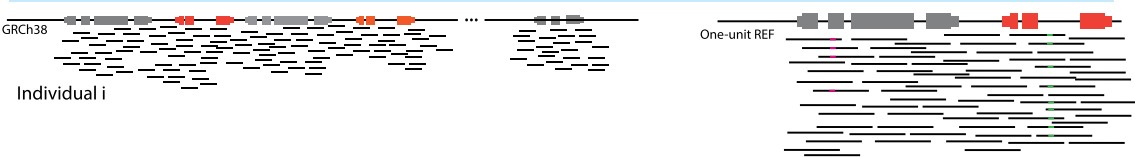

(3) Ascertain and estimate PSV allele fractions for all common sequence variants

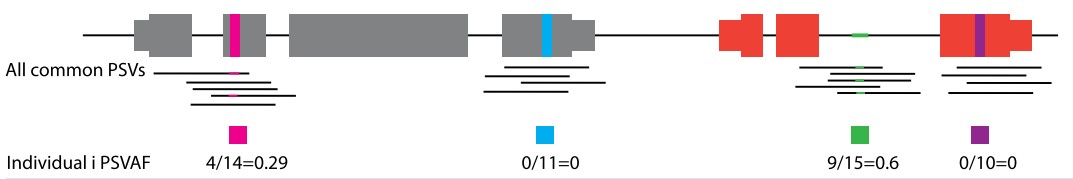

All common PSVs

Individual i PSVAF          4/14=0.29          0/11=0          9/15=0.6          0/10=0

(4) Convert PSV allele fractions to absolute copy number estimates

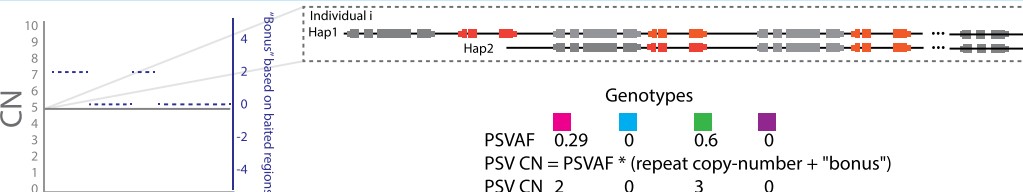

Genotypes

| | | | | |
|---|---|---|---|---|
| PSVAF | 0.29 | 0 | 0.6 | 0 |
| PSV CN = PSVAF * (repeat copy-number + "bonus") | | | | |
| PSV CN | 2 | 0 | 3 | 0 |

**Extended Data Fig. 4 | Overview of copy-number estimation for paralogous sequence variants (PSVs).** This figure provides a graphical overview of the pipeline we used to estimate copy-numbers of PSVs—that is, SNPs and indels carried on one or more copies of a multi-copy segment—from WGS read alignments (Supplementary Note, Section 9). We then refined these estimates using haplotype-sharing information.

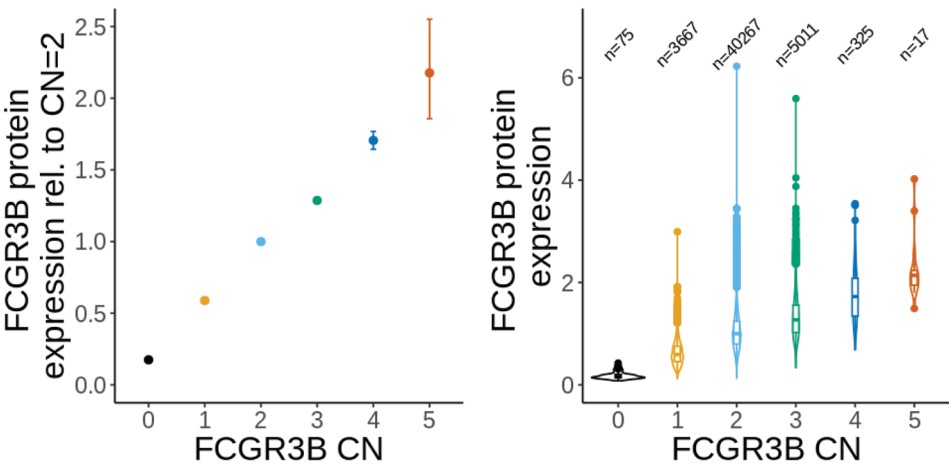

## (a) Proteomic validation of *FCGR3B* copy number

## (b) WES-WGS correlation of *FCGR3B* copy number

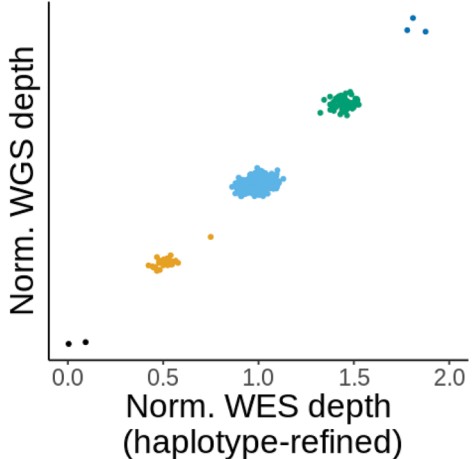

**Extended Data Fig. 5 | Validation of *FCGR3B* copy number estimation.**
**a**, Left, normalized protein expression of FCGR3B for each *FCGR3B* copy-number state relative to CN = 2. Estimates and 95% confidence intervals were obtained from linear regression analyses of NPX values and then converted to the linear scale ($2^{NPX}$). Right, distribution of normalized protein expression of FCGR3B converted to the linear scale ($2^{NPX}$) for each *FCGR3B* copy-number state. Counts of individuals with each copy-number state are shown above the corresponding violin. Boxplots display median value (center line), hinges denote first and third quartile (25th and 75th percentile), and whiskers extend from upper (resp. lower) hinge to the largest (resp. smallest) value at most 1.5 times the interquartile range away from the hinge; all other points are considered outliers and plotted individually. **b**, Scatter plot of normalized WGS and WES read depths at *FCGR3B* for 500 UKB participants. Points are colored based on the estimated *FCGR3B* copy number derived from WES.

# Reporting Summary

## Statistics

For all statistical analyses, confirm that the following items are present in the figure legend, table legend, main text, or Methods section.

| n/a | Confirmed | |
|---|---|---|
| ☐ | ☒ | The exact sample size (*n*) for each experimental group/condition, given as a discrete number and unit of measurement |
| ☐ | ☒ | A statement on whether measurements were taken from distinct samples or whether the same sample was measured repeatedly |
| ☐ | ☒ | The statistical test(s) used AND whether they are one- or two-sided *Only common tests should be described solely by name; describe more complex techniques in the Methods section.* |
| ☐ | ☒ | A description of all covariates tested |
| ☐ | ☒ | A description of any assumptions or corrections, such as tests of normality and adjustment for multiple comparisons |
| ☐ | ☒ | A full description of the statistical parameters including central tendency (e.g. means) or other basic estimates (e.g. regression coefficient) AND variation (e.g. standard deviation) or associated estimates of uncertainty (e.g. confidence intervals) |
| ☐ | ☒ | For null hypothesis testing, the test statistic (e.g. *F*, *t*, *r*) with confidence intervals, effect sizes, degrees of freedom and *P* value noted *Give P values as exact values whenever suitable.* |
| ☒ | ☐ | For Bayesian analysis, information on the choice of priors and Markov chain Monte Carlo settings |
| ☒ | ☐ | For hierarchical and complex designs, identification of the appropriate level for tests and full reporting of outcomes |
| ☐ | ☒ | Estimates of effect sizes (e.g. Cohen's *d*, Pearson's *r*), indicating how they were calculated |

*Our web collection on statistics for biologists contains articles on many of the points above.*

## Software and code

Policy information about availability of computer code

| Data collection | None |
|---|---|
| Data analysis | Custom code used to perform haplotype-informed CNV analysis of UKB WES data has been deposited at Zenodo (10.5281/zenodo.10529671). The following open-source software packages were also used: samtools v1.11, mosdepth v0.2.5, bedtools v2.27.1, BLAT v35, BWA v0.7.17, HTSbox r345, plink v1.9, plink v2.0, BOLT-LMM v2.4.1, REGENIE v2.2.4, SuSiE v0.12.27, R (3.6.3). |

For manuscripts utilizing custom algorithms or software that are central to the research but not yet described in published literature, software must be made available to editors and reviewers. We strongly encourage code deposition in a community repository (e.g. GitHub). See the Nature Portfolio guidelines for submitting code & software for further information.

## Data

Policy information about availability of data

All manuscripts must include a data availability statement. This statement should provide the following information, where applicable:
- Accession codes, unique identifiers, or web links for publicly available datasets
- A description of any restrictions on data availability
- For clinical datasets or third party data, please ensure that the statement adheres to our policy

Individual-level CNV calls and continuous-valued estimates of relative copy number in UKB will be returned to UK Biobank. Summary statistics for CNV-phenotype association tests are available at https://data.broadinstitute.org/lohlab/UKB_WES_CNV_sumstats/ and have been deposited at Zenodo (10.5281/

## Research involving human participants, their data, or biological material

Policy information about studies with underline{human participants or human data}. See also policy information about underline{sex, gender (identity/presentation), and sexual orientation} and underline{race, ethnicity and racism}.

| | |
|---|---|
| Reporting on sex and gender | We included genetically-determined biological sex as a covariate in analyses. |
| Reporting on race, ethnicity, or other socially relevant groupings | We used self-reported ethnic background (UK Biobank Data-Field 21000) to define the primary analysis set and to compare allele frequencies across ethnic groupings. |
| Population characteristics | Prospective cohort study (~500,000 individuals from across the United Kingdom); individuals were between 40 and 69 years old at recruitment (Sudlow et al. 2015 PLOS Medicine). |
| Recruitment | Recruitment into UK Biobank has been described previously (Sudlow et al. 2015 PLOS Medicine). |
| Ethics oversight | Ethics approval for the UK Biobank study was obtained from the North West Centre for Research Ethics Committee (Bycroft et al. 2018 Nature). The present study analyzed de-identified data previously collected by UK Biobank and did not require additional ethics oversight. |

Note that full information on the approval of the study protocol must also be provided in the manuscript.

# Field-specific reporting

Please select the one below that is the best fit for your research. If you are not sure, read the appropriate sections before making your selection.

☒ Life sciences  ☐ Behavioural & social sciences  ☐ Ecological, evolutionary & environmental sciences

For a reference copy of the document with all sections, see nature.com/documents/nr-reporting-summary-flat.pdf

# Life sciences study design

All studies must disclose on these points even when the disclosure is negative.

| | |
|---|---|
| Sample size | We conducted genetic association analyses on 454,682 individuals (all UK Biobank participants of self-reported White ethnicity who were not excluded by one of the filters below). |
| Data exclusions | We excluded individuals with trisomy 21, blood cancer, aberrantly many CNV calls, and those who had withdrawn at the time of our study. |
| Replication | We analyzed the All of Us and BioBank Japan data sets to replicate the key associations identified in UK BIobank; all key associations replicated. |
| Randomization | Not applicable to our study; participants were analyzed together and not allocated into groups. |
| Blinding | Not applicable to our study; all data were previously collected, and participants were not allocated into groups. |

# Reporting for specific materials, systems and methods

We require information from authors about some types of materials, experimental systems and methods used in many studies. Here, indicate whether each material, system or method listed is relevant to your study. If you are not sure if a list item applies to your research, read the appropriate section before selecting a response.

### Materials & experimental systems

| n/a | Involved in the study |
|---|---|
| ☒ ☐ | Antibodies |
| ☒ ☐ | Eukaryotic cell lines |
| ☒ ☐ | Palaeontology and archaeology |
| ☒ ☐ | Animals and other organisms |
| ☒ ☐ | Clinical data |
| ☒ ☐ | Dual use research of concern |
| ☒ ☐ | Plants |

### Methods

| n/a | Involved in the study |
|---|---|
| ☒ ☐ | ChIP-seq |
| ☒ ☐ | Flow cytometry |
| ☒ ☐ | MRI-based neuroimaging |

