## [Peer Review File · Nature Genetics]

Peer Review Information

Manuscript Title: Protein-altering variants at copy number variable regions influence diverse human phenotypes

Corresponding author name(s): Dr Margaux (L.A.) Hujoel, Dr Po-Ru Loh

Editorial Notes:

Transferred manuscripts This document only contains reviewer comments, rebuttal and decision letters for versions considered at Nature Genetics.

Reviewer Comments & Decisions:

Decision Letter, initial version:

28th September 2023

Dear Po-Ru,

Your Article "Hidden protein-altering variants influence diverse human phenotypes" has been seen by two referees. You will see from their comments below that, while they find your work of interest, Reviewer #1 has requested a few revisions to improve aspects of the presentation. We are interested in publishing your study in Nature Genetics, but we would like to see your response to these points in the form of a revised manuscript before we make a final decision and provide further formatting instructions.

To guide the scope of the revisions, the editors discuss the referee reports in detail within the team with a view to identifying key priorities that should be addressed in revision. In this case, we ask that you revise the presentation taking into account Reviewer #1's comments and suggestions. We hope you will find this prioritized set of referee points to be useful when revising your study. Please do not hesitate to get in touch if you would like to discuss these issues further.

We therefore invite you to revise your manuscript taking into account all reviewer and editor comments. Please highlight all changes in the manuscript text file. At this stage, we will need you to upload a copy of the manuscript in MS Word .docx or similar editable format.

We are committed to providing a fair and constructive peer-review process. Do not hesitate to contact us if there are specific requests from the reviewers that you believe are technically impossible or

unlikely to yield a meaningful outcome.

*2) If you have not done so already, please begin to revise your manuscript so that it conforms to our Article format instructions, available [here](http://www.nature.com/ng/authors/article_types/index.html). Refer also to any guidelines provided in this letter.

[redacted]

We hope to receive your revised manuscript within 4-8 weeks. If you cannot send it within this time, please let us know.

Nature Genetics is committed to improving transparency in authorship. As part of our efforts in this direction, we are now requesting that all authors identified as 'corresponding author' on published papers create and link their Open Researcher and Contributor Identifier (ORCID) with their account on the Manuscript Tracking System (MTS), prior to acceptance. ORCID helps the scientific community achieve unambiguous attribution of all scholarly contributions. You can create and link your ORCID from the home page of the MTS by clicking on 'Modify my Springer Nature account'. For more information, please visit www.springernature.com/orcid.

Sincerely,
Kyle

Kyle Vogan, PhD
Senior Editor
Nature Genetics
<https://orcid.org/0000-0001-9565-9665>

Referee expertise:

Referee #1: Genetics, structural variation, immune-mediated diseases

Referee #2: Genomics, structural variation, bioinformatics

Reviewers' Comments:

Reviewer #1:
Remarks to the Author:

This paper describes an analysis of copy number variation association with a range of phenotypes using the whole exome sequence data from UK Biobank. Previous publications have used similar sequence read depth approaches on the first releases of exome data to infer copy number and associate with disease (e.g. Fitzgerald and Birney). The difference in this study is that the authors use information from flanking SNP haplotypes to refine the CNV calling. This will have limitations but will enrich for CNVs that are identical by descent – or nearly identical by descent – and therefore have occurred on a single SNP haplotype. The result is that this approach has improved detection for rare and smaller CNVs. The authors then use these data in two ways – association with 57 quantitative traits and incorporating into gene burden tests.

The authors highlight a few interesting examples, including associations with challenging multiallelic loci at DEFA1A3 and FCGR3B/FCGR3A. The complexities of these loci are acknowledged and dealt with in a very reasonable way such that the associations are trustworthy.

Overall, the paper is a thorough analysis and shows interesting data that is of broad importance to human genetics and is a nice body of work. While further whole genome data from short and long read sequencing on UKBB will yield further associations and will retrospectively validate the robustness of the CNV calling shown here, the authors have been careful and thorough in ensuring the high quality of their results given the WES data provided.

I have a few relatively minor points the authors should address:

1. The terminology should be clarified and made consistent in the manuscript for readers not in the field. At the moment, the abstract and start of the introduction uses “structural variants and “SVs” but then transitions to CNVs and, occasionally, copy-number-polymorphisms, in the rest of the manuscript

and additional material. My suggestion would be to use CNV and copy number variation throughout, as this is justified as this is what you are measuring using read depth. SV can be mentioned at the start if needed, stating that CNV is a subset of SV.

2. To support the statement on page 9 "FCGR3B deletion has previously been associated with several autoimmune disorders" the Fanciulli et al paper has been cited (ref 41). I suggest that the authors re-read the paper, as it is based on a small dataset using a noisy assay and has not been replicated since it was published 16 years ago. A more recent paper that shows association of FCGR3B deletion with rheumatoid arthritis using a much larger dataset, and attempts, with partial success, to address the problems with analysing this locus, is Rahbari et al PMID: 27995740, and I'd suggest that the authors consider citing that paper instead.

3. For DEFA1/A3 locus, since the two genes differ only by one variant, the correct gene name is not DEFA1/A3 but DEFA1A3.

4. In the supplementary material, the descriptor "White British ancestry" is used. Using assumed skin colour is inaccurate, and "British" is not without political baggage... This should be replaced by something more accurate and neutral stating how these were defined – something like "UK individuals of recent European ancestries, as defined by PC clustering with 1000 Genome CEU samples", depending, obviously, on how it was done.

5. Errors/incomplete information in the reference list for the supplementary information – e.g., references missing volume and page numbers. Please check each citation carefully to ensure that the full information is provided.

6. Extended Data Figure 5 – it is unclear why the points are plotted in different shades of brown – there is no legend. If these indicate different copy numbers, then they should be clearly plotted in accessible colours (i.e. with strong contrast against white, color-blind friendly). For (a) the full spread of the data (i.e. individual points in a beeswarm plot) should be shown rather than mean and 95% CI of the mean. This would also show the relative numbers in each copy number category.

Reviewer #2:

Remarks to the Author:

In this manuscript, the authors present results from a new sophisticated method to measure copy number variation in 100,000's of exome datasets, both inside and outside segmental duplications. This enables the authors to identify a number of new associations that could not be seen from small variants. The manuscript is well-written, and the authors appropriately outline the limitations of their approach while highlighting its strengths, and I have no major suggestions for improvement.

Author Rebuttal to Initial comments

Response to reviews of NG-A62862-T (Hujoel et al.):**Reviewer #1:**

This paper describes an analysis of copy number variation association with a range of phenotypes using the whole exome sequence data from UK Biobank. Previous publications have used similar sequence read depth approaches on the first releases of exome data to infer copy number and associate with disease (e.g. Fitzgerald and Birney). The difference in this study is that the authors use information from flanking SNP haplotypes to refine the CNV calling. This will have limitations but will enrich for CNVs that are identical by descent – or nearly identical by descent – and therefore have occurred on a single SNP haplotype. The result is that this approach has improved detection for rare and smaller CNVs. The authors then use these data in two ways – association with 57 quantitative traits and incorporating into gene burden tests.

The authors highlight a few interesting examples, including associations with challenging multiallelic loci at DEFA1A3 and FCGR3B/FCGR3A. The complexities of these loci are acknowledged and dealt with in a very reasonable way such that the associations are trustworthy.

Overall, the paper is a thorough analysis and shows interesting data that is of broad importance to human genetics and is a nice body of work. While further whole genome data from short and long read sequencing on UKBB will yield further associations and will retrospectively validate the robustness of the CNV calling shown here, the authors have been careful and thorough in ensuring the high quality of their results given the WES data provided.

We appreciate these encouraging comments and the helpful suggestions below, which we have incorporated into the revised manuscript (as detailed below).

I have a few relatively minor points the authors should address:

1. The terminology should be clarified and made consistent in the manuscript for readers not in the field. At the moment, the abstract and start of the introduction uses “structural variants and “SVs” but then transitions to CNVs and, occasionally, copy-number-polymorphisms, in the rest of the manuscript and additional material. My suggestion would be to use CNV and copy number variation throughout, as this is justified as this is what you are measuring using read depth. SV can be mentioned at the start if needed, stating that CNV is a subset of SV.

We thank the reviewer for noticing this inconsistent use of terminology, which we agree could cause confusion. As suggested, we have revised the manuscript to use CNV and copy number variation throughout, only mentioning SV at the start and stating that CNV is a subset of SV.

2. To support the statement on page 9 “FCGR3B deletion has previously been associated with several autoimmune disorders” the Fanciulli et al paper has been cited (ref 41). I suggest that the authors re-read the paper, as it is based on a small dataset using a noisy assay and has not been replicated since it was published 16 years ago. A more recent paper that shows association of FCGR3B deletion with rheumatoid arthritis using a much larger dataset, and attempts, with partial success, to address the problems with analysing this locus, is Rahbari et al PMID: 27995740, and I’d suggest that the authors consider citing that paper instead.

We thank the reviewer for calling this to our attention and pointing us to more recent, better-powered literature. We have replaced the Fanciulli et al. citation with Rahbari et al. (the new ref. 41) as suggested.

3. For DEFA1/A3 locus, since the two genes differ only by one variant, the correct gene name is not DEFA1/A3 but DEFA1A3.

We appreciate this correction and have changed all occurrences of *DEFA1/A3* to *DEFA1A3*.

4. In the supplementary material, the descriptor “White British ancestry” is used. Using assumed skin colour is inaccurate, and “British” is not without political baggage... This should be replaced by something more accurate and neutral stating how these were defined – something like “UK individuals of recent European ancestries, as defined by PC clustering with 1000 Genome CEU samples”, depending, obviously, on how it was done.

We agree that the descriptor “White British ancestry” is confusing and not ideal. This descriptor was actually first defined in the flagship UK Biobank paper (Bycroft et al. (2018) *Nature*) to describe a subset of individuals who self-reported “White British” ethnicity and had very similar genetic ancestry based on principal component analysis. It subsequently became a widely-used data field (“in.white.British.ancestry.subset”) in the main UK Biobank genotyping data release (Resource 531, <https://biobank.ctsu.ox.ac.uk/showcase/refer.cgi?id=531>).

To address this issue while facilitating reproducibility, we have replaced the descriptor with the name of the data field (placing it within quotation marks and formatting it as it appears in the data release) and have indicated how the sample subset was actually defined:

“... we restricted reference samples to the 409K “in.white.British.ancestry.subset” (a previously-defined subset of UK Biobank participants who self-reported White British ethnicity and had very similar genetic ancestry based on principal component analysis) and we further excluded related samples (Bycroft et al., 2018).”

5. Errors/incomplete information in the reference list for the supplementary information – e.g., references missing volume and page numbers. Please check each citation carefully to ensure that the full information is provided.

Thanks; we have updated the citations in the supplementary text to include the full information.

6. Extended Data Figure 5 – it is unclear why the points are plotted in different shades of brown – there is no legend. If these indicate different copy numbers, then they should be clearly plotted in accessible colours (i.e. with strong contrast against white, color-blind friendly). For (a) the full spread of the data (i.e. individual points in a beeswarm plot) should be shown rather than mean and 95% CI of the mean. This would also show the relative numbers in each copy number category.

We appreciate these helpful suggestions for improving EDF 5 and have incorporated them in the revised figure. The colors in the figure do indeed indicate different copy numbers, which we have now clarified in the legend. We have also changed the color scheme to be color-blind friendly. As there were too many individual measurements for a beeswarm plot (e.g., 40,267 CN=2 individuals), we instead added a violin plot, as we agree that showing the distribution of the data is informative. We have also indicated the count of individuals with each copy number in the plot.

Reviewer #2:

In this manuscript, the authors present results from a new sophisticated method to measure copy number variation in 100,000's of exome datasets, both inside and outside segmental duplications. This enables the authors to identify a number of new associations that could not be seen from small variants. The manuscript is well-written, and the authors appropriately outline the limitations of their approach while highlighting its strengths, and I have no major suggestions for improvement.

We appreciate these kind comments.

Decision Letter, first revision:

21st November 2023

Dear Po-Ru,

Thank you for submitting your revised manuscript "Hidden protein-altering variants influence diverse human phenotypes" (NG-A62862R). In light of the positive referee feedback at the initial round of review and your responses to the referees' comments, we will be happy in principle to publish your study in Nature Genetics as an Article pending final revisions to comply with our editorial and formatting guidelines.

We are now performing detailed checks on your paper, and we will send you a checklist detailing our editorial and formatting requirements soon. Please do not upload the final materials or make any revisions until you receive this additional information from us.

Thank you again for your interest in Nature Genetics. Please do not hesitate to contact me if you have any questions.

Sincerely,
Kyle

Kyle Vogan, PhD
Senior Editor
Nature Genetics
<https://orcid.org/0000-0001-9565-9665>

Final Decision Letter:

8th February 2024

Dear Po-Ru,

I am delighted to say that your manuscript "Protein-altering variants at copy number variable regions influence diverse human phenotypes" has been accepted for publication in an upcoming issue of Nature Genetics.

Over the next few weeks, your paper will be copyedited to ensure that it conforms to Nature Genetics style. Once your paper is typeset, you will receive an email with a link to choose the appropriate

publishing options for your paper and our Author Services team will be in touch regarding any additional information that may be required.

Your paper will be published online after we receive your corrections and will appear in print in the next available issue. You can find out your date of online publication by contacting the Nature Press Office (press@nature.com) after sending your e-proof corrections.

Before your paper is published online, we will be distributing a press release to news organizations worldwide, which may very well include details of your work. We are happy for your institution or funding agency to prepare its own press release, but it must mention the embargo date and Nature Genetics. Our Press Office may contact you closer to the time of publication, but if you or your Press Office have any enquiries in the meantime, please contact press@nature.com.

Please note that Nature Genetics is a Transformative Journal (TJ). Authors may publish their research with us through the traditional subscription access route or make their paper immediately open access through payment of an article-processing charge (APC). Authors will not be required to make a final decision about access to their article until it has been accepted. Find out more about Transformative Journals

Authors may need to take specific actions to achieve compliance with funder and institutional open access mandates. If your research is supported by a funder that requires immediate open access (e.g. according to Plan S principles) then you should select the gold OA route, and we will direct you to the compliant route where possible. For authors selecting the subscription publication route, the journal's standard licensing terms will need to be accepted, including . Please let your coauthors and your institutions' public affairs office know that they are also welcome to order reprints by this method.

If you have not already done so, we invite you to upload the step-by-step protocols used in this manuscript to the Protocols Exchange, part of our on-line web resource, natureprotocols.com. If you complete the upload by the time you receive your manuscript proofs, we can insert links in your article that lead directly to the protocol details. Your protocol will be made freely available upon publication of your paper. By participating in natureprotocols.com, you are enabling researchers to more readily reproduce or adapt the methodology you use. [Natureprotocols.com](http://natureprotocols.com) is fully searchable, providing your protocols and paper with increased utility and visibility. Please submit your protocol to <https://protocolexchange.researchsquare.com/>. After entering your [nature.com](http://www.nature.com) username and password you will need to enter your manuscript number (NG-A62862R1). Further information can be found at <https://www.nature.com/nature-portfolio/editorial-policies/reporting-standards#protocols>

Sincerely,
Kyle

Kyle Vogan, PhD
Senior Editor
Nature Genetics
<https://orcid.org/0000-0001-9565-9665>